# Components of 21 years (1995-2015) of Absolute Sea Level Trends in the Arctic

Carsten Bjerre Ludwigsen[1], Ole Baltazar Andersen[1], and Stine Kildegaard Rose[1]

[1]DTU Space, Elektrovej 328, 2800 Kgs. Lyngby, Denmark

**Correspondence:** Carsten Bjerre Ludwigsen (caanlu@space.dtu.dk)

**Abstract.** The Arctic Ocean is at the frontier of the fast changing climate in the northern latitudes and sea level trends is a bulk measure of ongoing processes related climate change. Observations of sea level in the Arctic Ocean are nonetheless difficult to validate with independent measurements and is globally the region where the sea level trend (SLT) is most uncertain. The aim of this study is to create an satellite-independent reconstruction of Arctic SLT, as it is observed by altimetry and tide gauges (TG).

Previous studies uses the Gravity Recovery and Climate Experiment (GRACE) observations to estimate the manometric (mass component of) SLT. GRACE estimates are however challenged by large mass changes on land, which are difficult to seperate from much smaller ocean mass changes. Furthermore is GRACE not available before 2003 which significantly limits the period and makes the trend more vulnerable to short-term changes. As an alternative approach, this study estimates the climate change driven Arctic manometric SLT from the Arctic sea level fingerprints of glaciers, Greenland, Antarctica and Glacial Isostatic

Adjustment (GIA) and adding the long-term Inverse Barometer (IB) effect. The halo- and thermosteric components completes the reconstructed Arctic SLT and are estimated by interpolating 300,000 temperature (T) and salinity (S) in-situ observations.

     The SLT from 1995-2015 is compared to the observed SLT from altimetry and 12 selected tide gauges (TG) corrected for vertical land movement (VLM). The reconstructed estimate manifests that the salinity-driven halosteric component is dominating the spatial SLT-pattern with variations between -7 and 10 mm y$^{-1}$. The manometric SLT is in comparison estimated

to 1-2 mm y$^{-1}$ for most of the Arctic Ocean. The reconstructed SLT shows a larger sea level rise in the Beaufort Sea compared to altimetry. An issue that is also identified by previous studies. A TG-observed sea level rise in the Siberian Arctic is in contrast to the sea level fall from the reconstructed and altimetric estimate.

     From 1995-2015 does the reconstructed SLT agree within the 68% confidence interval with the SLT from observed altimetry in 87% of the Arctic between 65N and 82N (R=0.50) and with 5 of 12 TG-derived (VLM corrected) SLT estimates. The

residuals are seemingly smaller than results from previous studies using GRACE-estimates and modelled T/S-data. The spatial correlation of the reconstructed SLT to altimetric SLT during the GRACE-period (2003-2015) is R=0.38 and R=0.34/R=0.37 if GRACE-estimates are used instead of the constructed manometric component. Thus is the reconstructed manometric component suggested as an legitimate alternative to GRACE, that can be projected into the past and future.

# 1 Introduction

The Arctic is globally the region with the fastest changing climate and is warming twice the rate of the global average (Box et al., 2019). The resulting enhanced deglaciation of land, decline of sea ice cover and ocean freshening has several affects on sea level. Hence are observations of sea level a measure of multiple ongoing processes, but naturally lacks information on the source of sea level change. Parallel are sea level observations from satellite altimetry and tide gauges of the Arctic Ocean challenged by an harsh environment, sea ice floes and lack of spatial coverage (Smith et al., 2019). Decomposing the observed long term sea level change provides insight into the regional effects of ongoing climate processes and helps consolidating the observed sea level.

Satellite altimetry has measured the sea level of the Arctic Ocean since 1991 with ESA's European Remote Sensing (ERS)-1 satellite being the first reaching polar latitudes. Laxon et al. (2003) were the first to study Arctic sea level from the ERS-1/2 satellites to produce sea ice thicknesses. Since then many have followed e.g. (Peacock and Laxon, 2004; Giles et al., 2012; Prandi et al., 2012; Cheng et al., 2015; Rose et al., 2019), but large variability in particular in sea ice-covered regions are still present (Armitage et al., 2016; Carret et al., 2017; Rose et al., 2019).

The sea level budget has been resolved on global and basin-wide scales for observations since the begin of the 19th century by using a combination of in-situ data, satellite observations and probabilistic analysis (Church and White, 2011a; WCRP, 2018; Dangendorf et al., 2019; Royston et al., 2020; Frederikse et al., 2020), but these studies are neglecting the polar regions due to large uncertainties and the relative small area of the Arctic Ocean in a global context.

Previous studies have made attempts to reconstruct sea level in the Arctic spatially (Henry et al., 2012; Carret et al., 2017; Raj et al., 2020; Ludwigsen and Andersen, 2020), while Armitage et al. (2016) estimates the mass and steric SLT-components as basin-wide average. All previous studies are using different solutions of GRACE to obtain their result. Henry et al. (2012) used CSR-RL04 (Bonin et al., 2012) from 2003-2009, Armitage et al. (2016) used JPL-RL05 (Chambers and Bonin, 2012) from 2003-2014, and Raj et al. (2020) used GSFC-mascons (RL05) (Luthcke et al., 2013) from 2003-2018. Carret et al. (2017) and Ludwigsen and Andersen (2020) compared the manometric sea level trend of different GRACE-solutions which revealed discrepancies of 5-10 mm y$^{-1}$ among GRACE trend-estimates in large areas of the Arctic. This disagreement has been attributed to different methods to remove contamination from land mass changes that leaks into the ocean signal observed by GRACE (Mu et al., 2020). Hence is the chosen GRACE-solution consequential for the closing of the sea level budget and its ability to validate altimetric observations.

In contrast to the mentioned Arctic sea level budget studies, this study bypasses GRACE-based ocean mass estimates by calculating the sea level fingerprints of contemporary land ice loss, glacial isostatic adjustment (GIA) and atmospheric pressure (inverse barometer, IB) which results in a long-term manometric sea level trend estimate. This approach gives three advantages over GRACE: (i) Insights of the different contributions to manometric sea level change, (ii) a longer time series that extends into the pre-GRACE era, which has the advantage, that non-secular and inter-annual ocean dynamic mass effects, which are mainly driven by the Arctic Oscillation (AO) (Henry et al., 2012; Volkov and Landerer, 2013; Peralta-Ferriz et al., 2014;

Armitage et al., 2018), are reduced and (iii) the mentioned problem of leakage from effects caused by the low spatial resolution (300-500 km (Tapley et al., 2004)) are avoided.

Combining the manometric 1995-2015 SLT estimates with satellite-independent steric SLT estimates (Ludwigsen and Andersen, 2020) aims to reconstruct the absolute SLT as it is observed by altimetry. Besides consolidating observed sea level change, the sea level budget decomposition permits analysis of the sources of contemporary long-term Arctic sea level change, which also aids predictions of future change.

## 2 Method

Sea level observations from satellite altimetry are measured relative to a terrestrial reference frame and is referred to as geocentric or absolute sea level (ASL) observations. Tide gauges (TG) measures the sea level while being grounded to the coast, and is affected by vertical deformations of the solid earth, called vertical land movement (VLM). When VLM is defined with respect to the same reference frame as altimetry and added to TG-measured relative sea level (RSL) the ASL is restored:

$$\text{ASL} = \text{RSL} + \text{VLM} \tag{1}$$

Changes of ASL ($\dot{\text{ASL}}$) originates either from changed ocean density (steric, $\dot{\eta}$) due to changes in salinity (halosteric) or temperature (thermosteric) or from changes in ocean mass, denoted as manometric sea level change, $\dot{\text{M}}$ (Gregory et al., 2019)). According to (Gregory et al., 2019), manometric sea level change can be referred to as the 'non-steric' sea level change and is assumed indifferent to the commonly used Ocean Bottom Pressure (OBP). In this study, the manometric component is both reconstructed (1995-2015) and retrieved from GRACE observations (2003-2015).

$$\dot{\text{ASL}} = \dot{\eta} + \dot{\text{M}} \tag{2}$$

As already mentioned, the steric sea level change is composed of halosteric ($\dot{\eta}_S$) and thermosteric ($\dot{\eta}_T$) sea level change:

$$\dot{\eta} = \dot{\eta}_S + \dot{\eta}_T \tag{3}$$

The manometric component is further divided into contributions from changes in the gravitational field, $G$ that together with a spatial uniform constant, $c$, composes the gravitational sea level fingerprint ($N$) due to different land-to-ocean mass changes, $i$, which in this study originates from either different sources of land ice (Greenland (GRE), Northern Hemisphere (NH) Glaciers and Antarctica (Ant) + Southern Hemisphere (SH) glaciers) or GIA. Change in atmospheric pressure (Inverse Barometer, IB) is added to the sea level fingerprints to create the total manometric sea level change, $\dot{\text{M}}$.

$$\dot{\text{M}} = \sum_i \dot{N}_i + \dot{\text{IB}} \quad , \text{ where } \quad \dot{N}_i = \dot{G}_i + \dot{c}_i \tag{4}$$

By substituting eq. 4 and eq. 3 into eq. 2, we achieve the reconstruction of absolute sea level, $\text{ASL}_r$, that is comparable with the altimetry observed ASL (denoted as $\text{ASL}_A$):

$$\dot{\text{ASL}}_r = \sum_i (\dot{G}_i + \dot{c}_i) + \dot{\text{IB}} + \dot{\eta}_S + \dot{\eta}_T \tag{5}$$

VLM is split into the viscoelastic solid earth deformation caused from past millennial ice (un-)loading, GIA, and the elastic adjustment from contemporary (1995-2015) change in ice loading, VLMe, which, as $G$, is a composite of the elastic response from different origins of land ice ($i$).

$$\dot{\text{VLM}} = \dot{\text{GIA}} + \sum \dot{\text{VLMe}}_i \tag{6}$$

Possible local VLMs not associated with glacial mass redistribution (i.e. non-glacial land water change, tectonics or oil depletion) is not accounted for since little knowledge on their VLM-contribution exist. Frederikse et al. (2019) estimated the non-glacial VLM from GRACE-observations to vary between -0.5 mm $\text{y}^{-1}$ in North America and +0.2 mm $\text{y}^{-1}$ in the Barents/Kara sea region.

Adding VLM (eq. 6) to TG-measured RSL, gives according to eq. 1 a third ASL estimate, $\text{ASL}_{\text{TG}}$:

$$\dot{\text{ASL}}_{\text{TG}} = \dot{\text{RSL}}_{\text{TG}} + \dot{\text{GIA}} + \sum \dot{\text{VLMe}}_i \tag{7}$$

## 3 Data

This study combines various in-situ data (temperature and salinity (T/S) profiles, tide gauges and ocean bottom recorders), satellite altimetry, GRACE-observations and model data (ECCOv4r4, VLM and geoid change) to reconstruct the Arctic sea level change. In this section follows a description of the different datasets and how they are obtained.

### 3.1 Altimetry

The DTU/TUM Arctic Ocean sea level anomaly (SLA) record (Rose et al., 2019) provides an independent estimate of ASL change ($\dot{\text{ASL}}_{\text{A}}$). The altimetric time series is covering the whole altimetric era given as monthly grids from September 1991 to September 2018, covering 65° N to 81.5°N and 180°W–179.5°E.

Geophysical corrections such as tides and atmospheric delays is applied to the altimetric sea level estimate. Leads (cracks in the sea ice cover) and open ocean are located and separated according to the different classification of their surfaces. The detection of leads is not flawless, and their sparse distribution in the sea ice cover, and the uncertainty of the the applied geophysical corrections in the Arctic (Stammer et al., 2014; Ricker et al., 2016) makes the sea level estimates more uncertain in the sea ice covered region. The altimetric record includes data from four ESA satellites: ERS-1 (1991-1995), ERS-2 (1995-2003), Envisat (2002-2010) and CryoSat-2 (2010-2018). It combines results of different retrackers as well as conventional and SAR-altimetry (Rose et al., 2019). In particular ERS-1/2 has a relatively low spatial resolution and measurements from leads in sea ice are limited. Observations are in particular sparse and uncertain in sea ice regions from ERS-1 (Rose et al., 2019), which is why the altimetry record used for this study begins in 1995. The SAR altimeter on CryoSat-2 is designed to measure over the sea ice cover, which increases the observations from leads and decreases the uncertainty (Rose et al., 2019). The applied version of the DTU/TUM altimetry product is not corrected for GIA or atmosphere pressure loading.

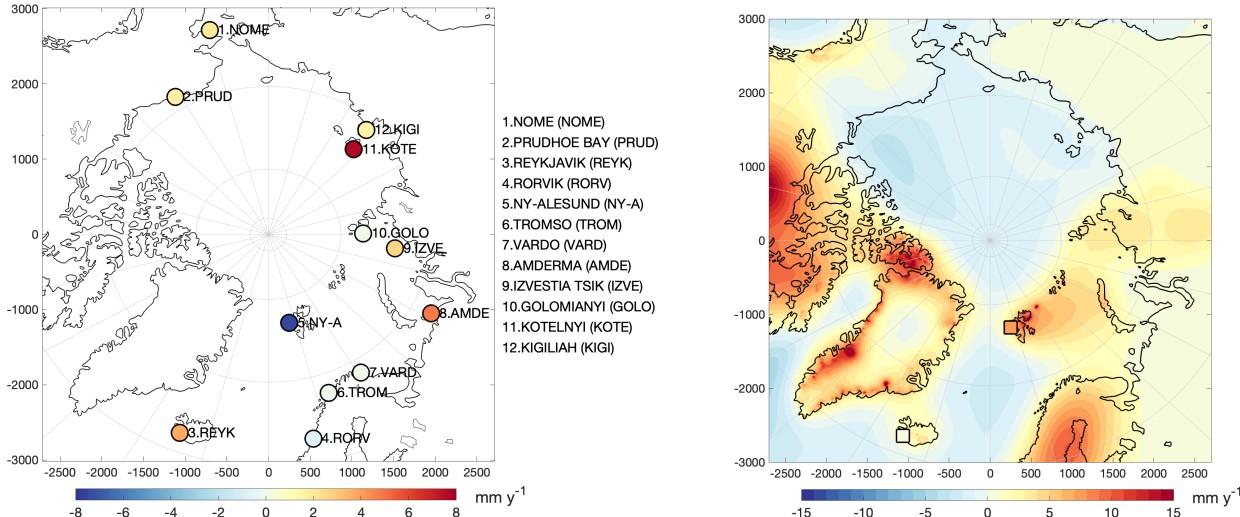

**Figure 1.** Left: 1995-2015 RSL trend [mm y$^{-1}$] and location of the selected tide gauges of this study. Right: 1995-2015 VLM-trend [mm y$^{-1}$] from the model of Ludwigsen et al. (2020b). The VLM-trend from the GNSS-sites at Reykjavik and Ny-Ålesund are shown with squared color coded markers.

## 3.2 Tide Gauges and Vertical Land Movement

Observations from tide gauges (TG) is obtained from the Permanent Service of Sea Level (PSMSL)-database (Holgate et al., 2012) given as monthly SLA. TGs with a consistent time series are few and unevenly distributed in the Arctic (Henry et al., 2012; Limkilde Svendsen et al., 2016). Usually, TG-observed RSL is aligned to ASL by utilizing vertical velocities from a nearby Global Navigation Satellite System (GNSS) receiver. However, only few reliable GNSS-data at the Arctic coast are spanning the time period of this study (Wöppelmann and Marcos, 2016; Ludwigsen et al., 2020a) and restricting TGs to locations with usable GNSS significantly limits the selection of TGs further. Therefore, an Arctic-wide VLM-model with annual VLM-rates from 1995-2015 (Ludwigsen et al., 2020a) is used as a substitute for GNSS (figure 1). A detailed comparison between vertical rates from the used VLM-model and GNSS-measurements (from URL6B (Santamaría-Gómez et al., 2017)) showed good agreement, in particular along the Norwegian Coast (Ludwigsen et al., 2020a).

The region around the Ny-Ålesund TG and Reykjavik TG experiences extraordinary VLM that is caused by substantial deglaciation during the Little Ice Age (LIA) (Svalbard) and low mantle viscosities in Iceland and Greenland. This is not captured in the spatially uniform REF6371 earth model (Kustowski et al., 2007) used in the VLM-model. Therefore, the two sites are corrected with nearby GNSS instead of the VLM-model. Large residual trends between the VLM-model (-1.4mm y$^{-1}$) and GNSS (-3.2 mm y$^{-1}$) was also found at Prudhoe Bay. This additional subsidence is likely caused by near-by construction or oil depletion sites. However, the tide gauge is located on a peninsula reaching into the Beaufort Sea 10 km away from the GNSS-location, which is why the VLM-model is trusted over the GNSS-measurement.

The VLM-model is composed from eq. 6. The GIA-component is based on the Caron2018 GIA-model (Caron et al., 2018), which includes an uncertainty estimate. Reported discrepancies from other GIA-models in central North America and Greenland (Caron et al., 2018; Ludwigsen et al., 2020a) has little affect at the locations of TGs of this study. Annual rates of VLMe is estimated from the 1995-2015 annual change of land ice using the Regional Elastic Rebound Calculator (REAR) (Melini et al., 2015). REAR also provides the gravitational response $G$ to land ice change used for estimating the manometric sea level. Uncertainties of the elastic VLM-estimates are mainly due to uncertainties of the applied land ice change. An additional 10% of the VLM-signal (after Wang et al. (2012)) is added to represent uncertainties associated with the REF6371 earth model (Kustowski et al., 2007) applied in REAR. The VLM contribution from non-tidal ocean loading (NOL) (van Dam et al., 2012) and rotational feedback (RF) (King et al., 2012) are in total of an order of $\pm 0.3$ mm y$^{-1}$ and are included in the VLM-contribution from Northern Hemisphere glaciers.

12 TGs are selected (geographical locations shown in figure 1) based on visual inspection of the monthly time series and to ensure that as many regions of the Arctic is represented as possible. 3-month averaged time series and linear trend of TG observed sea level (RSL$_{TG}$) and VLM-corrected sea level (ASL$_{TG}$) from 1995-2015 is shown in figure 2. The annual VLM-model is interpolated onto the TG time series and the linear trend is determined with least-squares method using months with available data between 1995 and 2015. In particular, the Alaskian and Siberian TGs have months with no or unreliable data (flagged by PSMSL). However, there is no evident seasonality in the missing months and therefore the trend estimates are not significantly affected by a seasonal bias.

Reykjavik (64.2°N), Nome (64.5°N), and Rorvik (64.9°N) are located off the edge of the altimetric data, which only extends to 65°N, but are nevertheless included to extend the spatial distribution of the TG-sites.

From figure 2, we see that the RSL-trends in the Arctic vary with nearly +/- 1 cm y$^{-1}$, with Ny-Ålesund on Svalbard having a negative RSL-trend of -7.45 mm y$^{-1}$, while Kostelnyi Island between the Laptev and East Siberian Sea shows a positive trend of 7.67 mm y$^{-1}$. However, after applying the VLM-correction, all TGs show a positive ASL-trend within a range of 0.38 mm $^{-1}$ (Prudhoe Bay) and 6.55 mm $^{-1}$ (Kostelnyi).

## 3.3 Steric sea level

The steric estimate is derived from the DTU Steric product (Ludwigsen and Andersen, 2020). The steric heights are calculated from a three dimensional T/S-grid that is interpolated from more than 300,000 T/S profiles and thus not constrained by any satellite observations. This approach is different to Morison et al. (2012) and Armitage et al. (2016), that use a difference between altimetry and GRACE to estimate steric heights and Henry et al. (2012); Carret et al. (2017); Raj et al. (2020), that use model-estimates of T/S to calculate the steric component.

T/S-profiles from buoys, ice-tethered profiles and ship expeditions in the Arctic Ocean are as shown in figure 3 spatially and temporally unevenly distributed and also depends on seasonal accessibility (Behrendt et al., 2017). Especially, in the shallow seas along the Siberian Coast (Ludwigsen and Andersen, 2020) is the data resolution poor and the areas with largest uncertainty. In the interior of the Arctic Ocean mostly summer data are available, while in the North Atlantic decent data coverage is reached year-around (figure 3). Temperature and salinity data are interpolated by kriging into a monthly 50x50 km

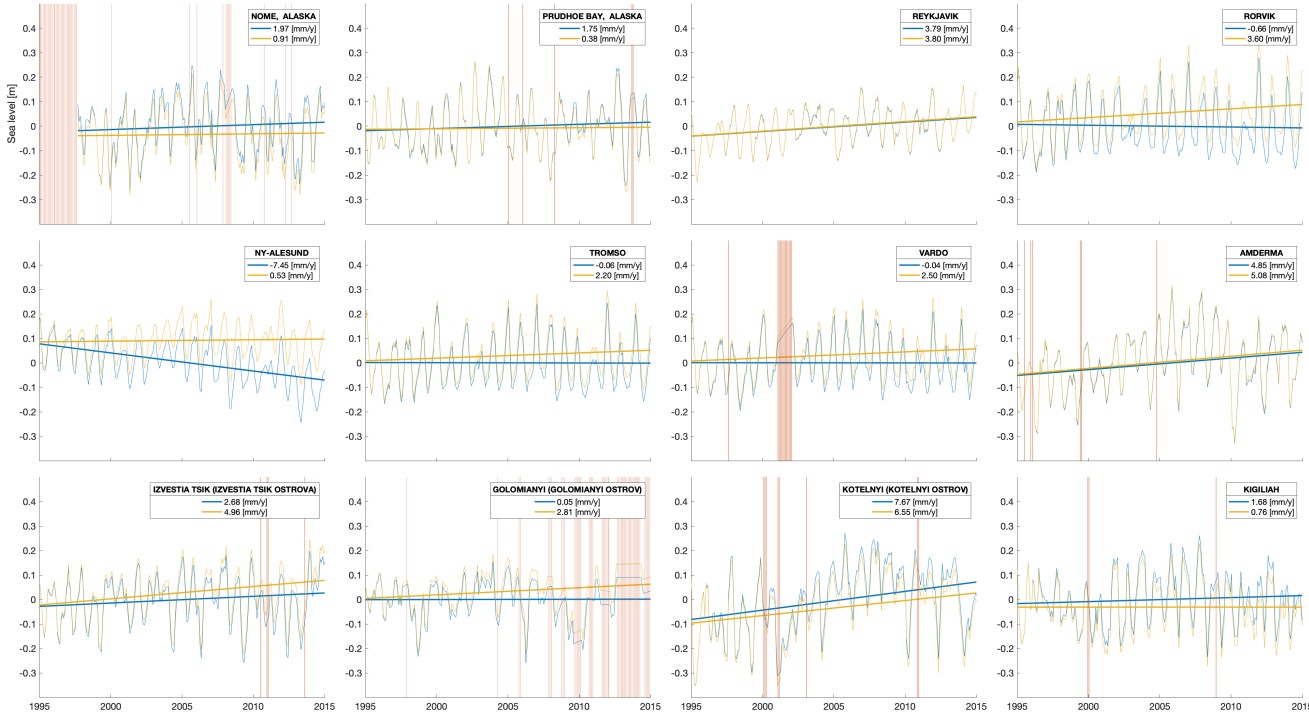

**Figure 2.** Relative sea level [m] from 1995-2015 registered at the 12 tide gauge from the PSMSL-database (Holgate et al., 2012)]. Blue line represents the 3-month running average, while the thick line is the linear trend (trend estimate [mm y$^{-1}$] shown in legend). Yellow line represents the absolute sea level and trend, equal to the blue line corrected for VLM with a VLM-model (Ludwigsen et al., 2020b) (except Ny-Ålesund and Reykjavik that are corrected with an extrapolated GNSS-trend). The vertical lines indicate where observations are missing and the sea level is linearly interpolated from adjacent months.

spatial grid on 41 depth levels. If values are more than $3\sigma$ away from the mean of neighbouring grid cells, values from the same month in adjacent years is used.

Following the notion of Gill and Niller (1973); Stammer (1997); Calafat et al. (2012); Ludwigsen and Andersen (2020), the change in steric sea level is calculated as the sum of halosteric sea level, $\eta_S$ and thermosteric sea level, $\eta_T$ (equation 3). From the depth profiles of the T/S grid, $\eta_S$ and $\eta_T$ are calculated:

$$\eta_S = -\frac{1}{\rho_0} \int_{-H}^{0} \beta S' dz \tag{8}$$

$$\eta_T = \frac{1}{\rho_0} \int_{-H}^{0} \alpha T' dz \tag{9}$$

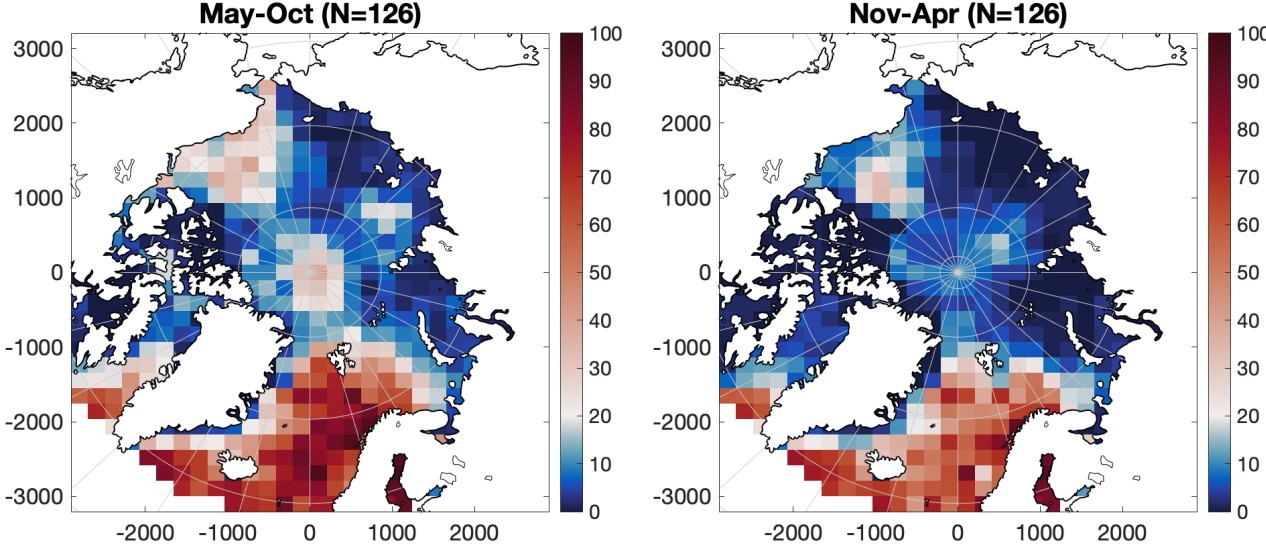

**Figure 3.** Percentage of months with available T/S data in 200x200 km grid cells. Left map: Summer months (May-October). Right map: Winter months (November-April).

where $H$ denotes the minimum height (maximum depth (z)). The maximum integration depth is as in Ludwigsen and Andersen (2020) 2000 meters. $S'$ and $T'$ are defining salinity and temperature anomalies, with reference values 0 C°and 35psu, respec-
tively. $\beta$ is the saline contraction coefficient and $\alpha$ is the thermal expansion coefficient. The opposite sign of $\eta_S$ is needed since $\beta$ represents a contraction (opposite to thermal expansion). $\alpha$ and $\beta$ are functions of absolute salinity, conservative temperature and pressure, and is determined with the freely available TEOS-10 software (Roquet et al., 2015). Sea level trends of $\eta_S$ and $\eta_T$ from 1995-2015 are shown in figure 4.

### 3.4    Manometric sea level contributions

Maps of the individual contributions from 1995-2015 to the manometric SLT (from equation 4) are shown in figure 5. The gravitational sea level fingerprint ($\dot{G}$) of contemporary land ice change (equation 4) is computed, similar to the elastic VLM-component, by solving the elastic greens functions with REAR (Melini et al., 2015). The geoid change from GIA is provided by the Caron2018-model (Caron et al., 2018).

     The sea level fingerprint of each component (figure 5a-d) is retrieved by adding the spatially invariant constant $c$ (global
mean sea level change) to the gravitational change. $c$ is equal to the individual components contribution to global mean sea level (given in brackets of figure 5) (Spada, 2017). Following Spada (2017), $c$ is defined as

$$c_i = -\frac{M_i \rho_w}{A_O} - \langle G_i - \mathrm{VLM}_i \rangle \tag{10}$$

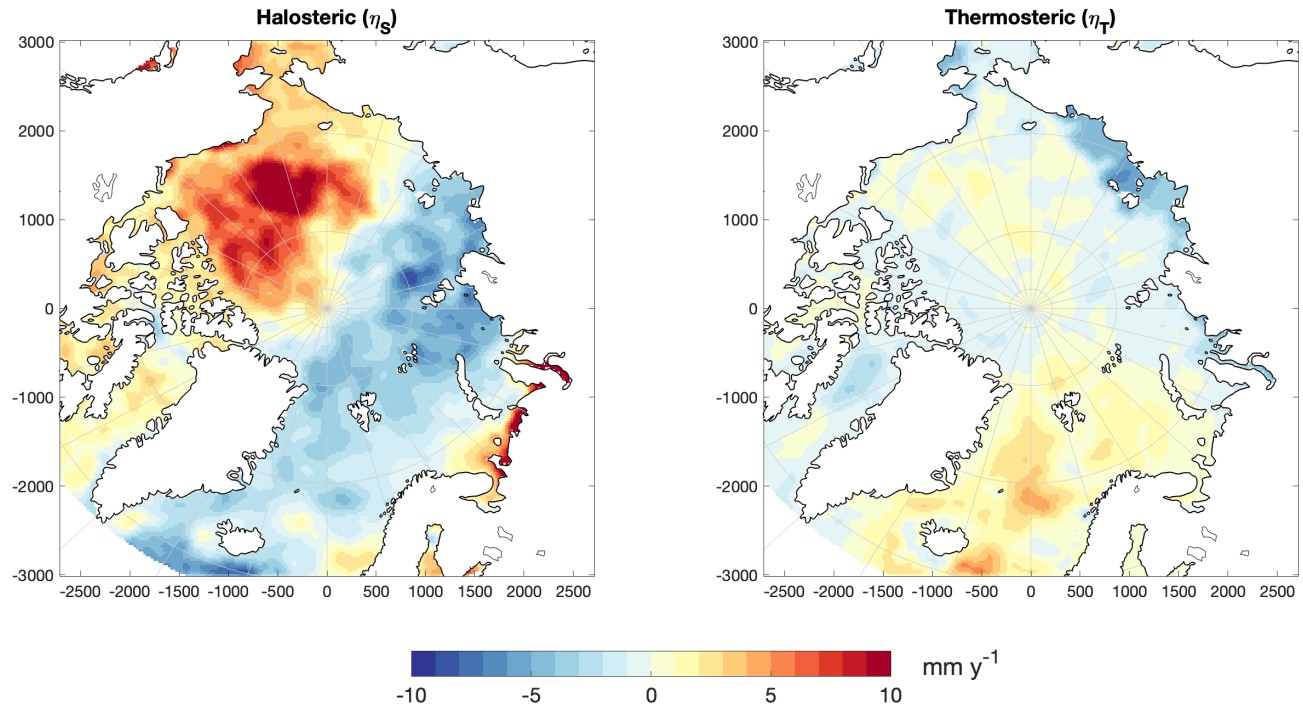

**Figure 4.** Halo- and thermosteric sea level trend [mm y$^{-1}$] from 1995-2015 derived from the DTU Steric sea level product (Ludwigsen and Andersen, 2020).

, where $M_i$ is the mass change of the ice model, $A_O$ is the total ocean area, $\rho_w$ is the average density of ocean water and $\langle ... \rangle$, denotes the average of the ocean surface. For calculating $c_i$, $G_i$ and VLM$_i$ for glaciers, individual glacial mass estimates are combined into a high resolution model for ice height change (Marzeion et al., 2012; Ludwigsen et al., 2020a). Models are used for mass loss estimates of Greenland (Khan et al., 2016) and Antarctica (Schröder et al., 2019). From 1995 to 2015, the estimated ice loss is 142 Gt y$^{-1}$ for Greenland, 206 Gt y$^{-1}$ for Northern Hemisphere glaciers and 105 Gt y$^{-1}$ for Antarctica and Southern Hemisphere glaciers, consistent with recent studies by Zemp et al. (2019) and Shepherd et al. (2018, 2020).

GIA is assumed to be unaffected by contemporary ice changes. This means that the GIA contribution to global mean sea level, $c$, is defined from the right part of equation 10, which is estimated to 0.3 mm y$^{-1}$ consistent with other studies (Peltier, 2009; Spada, 2017). The gravitational sea level change of RF and NOL is less than 0.05 mm y$^{-1}$, and are included in the Northern Hemisphere glacial contribution to $G$.

The manometric SLTs is completed with the loading from atmospheric pressure, IB (figure 5e). IB is estimated by the simple relationship derived from the hydro-static equation (Naeije et al., 2000; Pugh and Woodworth, 2014). Monthly averaged

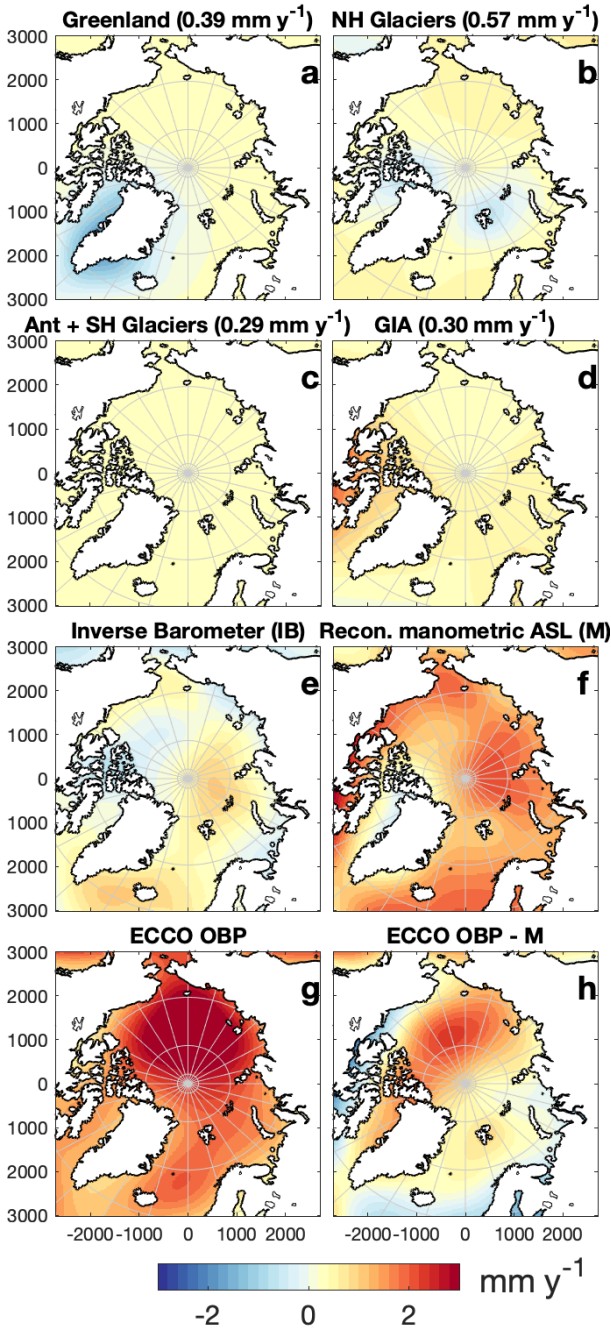

**Figure 5.** Contributions to the Arctic manometric sea level trend [mm y$^{-1}$] from 1995-2015. a-d shows $\dot{N}$ (eq. 4) for different sources of land-to-ocean mass changes with the global mean sea level contribution ($\dot{c}$) written in brackets: Greenland (incl. peripheral glaciers) (a), Northern Hemisphere (NH) glaciers (b), Antarctica (Ant) + Southern Hemisphere (SH) glaciers (c), and GIA (d). e shows the estimated Inverse Barometer trend. f shows the sum of a-e and hence the total reconstructed manometric sea level trend. Modelled OBP-estimate from ECCOv4r4 (Fukumori et al., 2019) (g). Difference between g and f (h).

pressure estimates from National Center for Environmental Prediction (NCEP) are used for surface pressure change $\Delta p$:

$$\text{IB} = -9.948 \, [\text{mm/mbar}] \, \Delta p \tag{11}$$

The total manometric SLTs ($\dot{\text{M}}$, figure 5f) is reconstructed as:

$$\dot{\text{M}} = \dot{N}_{\text{NHG}} + \dot{N}_{\text{GRE}} + \dot{N}_{\text{SH}} + \dot{N}_{\text{GIA}} + \dot{\text{IB}} \tag{12}$$

Figure 5g shows the OBP-trend from the ECCOv4r4-model (Estimating the Circulation and Climate of the Ocean (ECCO) version 4 release 4) (Forget et al., 2015; Fukumori et al., 2019), which is a model estimate of $\dot{\text{M}}$. The ECCO consortium (ecco-group.org) combines ocean circulation models with observations to estimate different physical parameters of the ocean. The model is among others constrained with observations from GRACE, satellite altimetry and in-situ T/S-profiles (Fukumori et al., 2019). The difference between ECCO OBP and $\dot{\text{M}}$ is displayed in figure 5h.

Figure 6 shows the reconstructed manometric SLT and ECCO OBP and two mascon solutions (release 06) of GRACE (JPL, (Watkins et al., 2015; Wiese et al., 2019) and GSFC (Loomis et al., 2019)) from 2003-2015 (different scale than figure 5) as well as the time series for three selected regions.

## 4   Results

The reconstructed SLT from 1995 to 2015 ($\dot{\text{ASL}}_r$) is shown in figure 7 panel (i) together with the SLT derived from altimetry (Rose et al., 2019). The residual of the reconstructed SLT to altimetry is shown in figure 9. In large, the spatial variability and residual is dominated by the halosteric sea level rise in the Beaufort Sea (10-15 mm y$^{-1}$), halosteric sea level fall in the East Siberian Sea (5-8 mm y$^{-1}$) and thermosteric sea level rise (2-5 mm y$^{-1}$) in the Norwegian Sea, where thermal expansion has a relatively larger impact compared to the near-freezing temperatures in the interior of the Arctic Ocean. A similar pattern is observed by altimetry (figure 7 panel (ii)), albeit a smaller sea level change in the Beaufort Sea and East Siberian Sea is detected.

The right panel of figure 9 shows the correlation matrix between $\dot{\text{ASL}}_{\text{A/TG}}$ and $\dot{\text{ASL}}_r$. The matrix shows that $\dot{\text{ASL}}_r$ and $\dot{\text{ASL}}_A$ are largely correlated (R=0.50). There is a large accumulation around 2 mm y$^{-1}$, with slightly higher $\dot{\text{ASL}}_A$ than $\dot{\text{ASL}}_r$. This originates from the underestimate of $\dot{\text{ASL}}_r$ (see figure map of 9) in the Norwegian Sea. This residual agrees with the ECCO OBP-$\dot{\text{M}}$ difference (figure 5h) and thus likely explained by the missing long-term dynamic sea level contribution of $\dot{\text{M}}$. From figure 9 also large residuals in the Beaufort Sea ($\dot{\text{ASL}}_r$ higher) and Siberian Coast ($\dot{\text{ASL}}_{\text{A/TG}}$ higher) are evident.

The sea level rise of the Beaufort Sea has been associated with a spin-up of the Beaufort Gyre from 2005 to 2010 that accumulated freshwater (Proshutinsky et al., 2009; Giles et al., 2012; Armitage et al., 2016). The halosteric trend in the Beaufort Sea and thermosteric trend in the Norwegian Sea is in agreement with the steric estimates from 1992-2014 by Carret et al. (2017) and from 2003-2016 by Raj et al. (2020). The steric-driven sea level fall in the East Siberian Sea is not recognized in extent and magnitude by these studies, but is nevertheless in agreement with the observed sea level fall by Armitage et al. (2016), which attributes this pattern to a rapid 10-15 cm fall in halosteric height in the East Siberian Seas from 2012-2014,

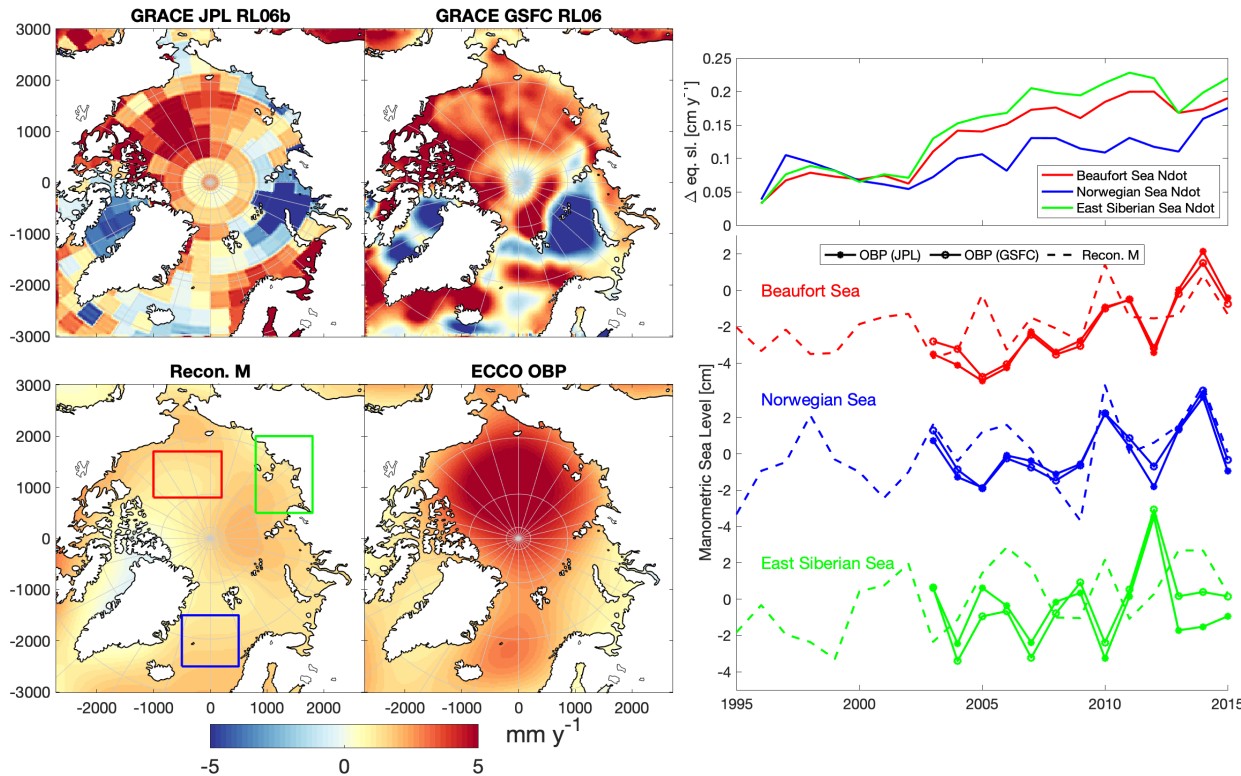

**Figure 6.** Left: Manometric (M) sea level/Ocean Bottom Pressure (OBP) trend estimates [mm y$^{-1}$] from 2003-2015 from two GRACE mascon products (top row), reconstructed from the contributions in figure 5 (bottom left) and from the ECCO-model (bottom right). Right: Botoom figure shows the yearly variation of GRACE OBP estimates and reconstructed manometric sea level in the Beaufort Sea region (red), Norwegian Sea (blue) and East SIberian Sea (green) as indicated by the squares in the map of the reconstructed manometric sea level [cm]. Top figure shows the year-to-year variation of $N$ (change in geoid + global sea level contribution, eq. 4) from all sources (land ice + GIA) (in sea level equivalent [cm y$^{-1}$]).

resulting in a 2003-2014 ASL trend of around -5 mm y$^{-1}$. The same 2012-2014 drop in East Siberian Sea is neither seen in ASL$_r$ or ASL$_A$, but is caused by a more general downward trend from 2002-2013 (figure 7 panel (iii)). Due to an assumed halosteric sea level high from 1998-2002 that coincides with an low in altimetry, is the correlation in the East Siberian Sea poor between ASL$_A$ and ASL$_r$ for the whole time series (R=-0.10), but improves when only considering the period from 2003-2015 235  (R=0.36). The large 2012-2014 steric sea level fall in the East Siberian Sea shown by Armitage et al. (2016) is also initiated by an apparent GRACE-observed manometric sea level rise seen in figure 7 panel (iii).

In the two other selected regions (Beaufort and Norwegian Sea) is the correlation between ASL$_A$ and ASL$_r$ better for the whole time series (1995-2015) than for the GRACE-period (2003-2015). In particular in the Beaufort Sea, is the correlation between ASL$_A$ and ASL$_r$ better before 2010. The correlation with altimetry is not significantly approved when ASL is 240  reconstructed using GRACE-estimates (ASL$_{r/grace}$ for the two regions (figure 7 panel (iii)).

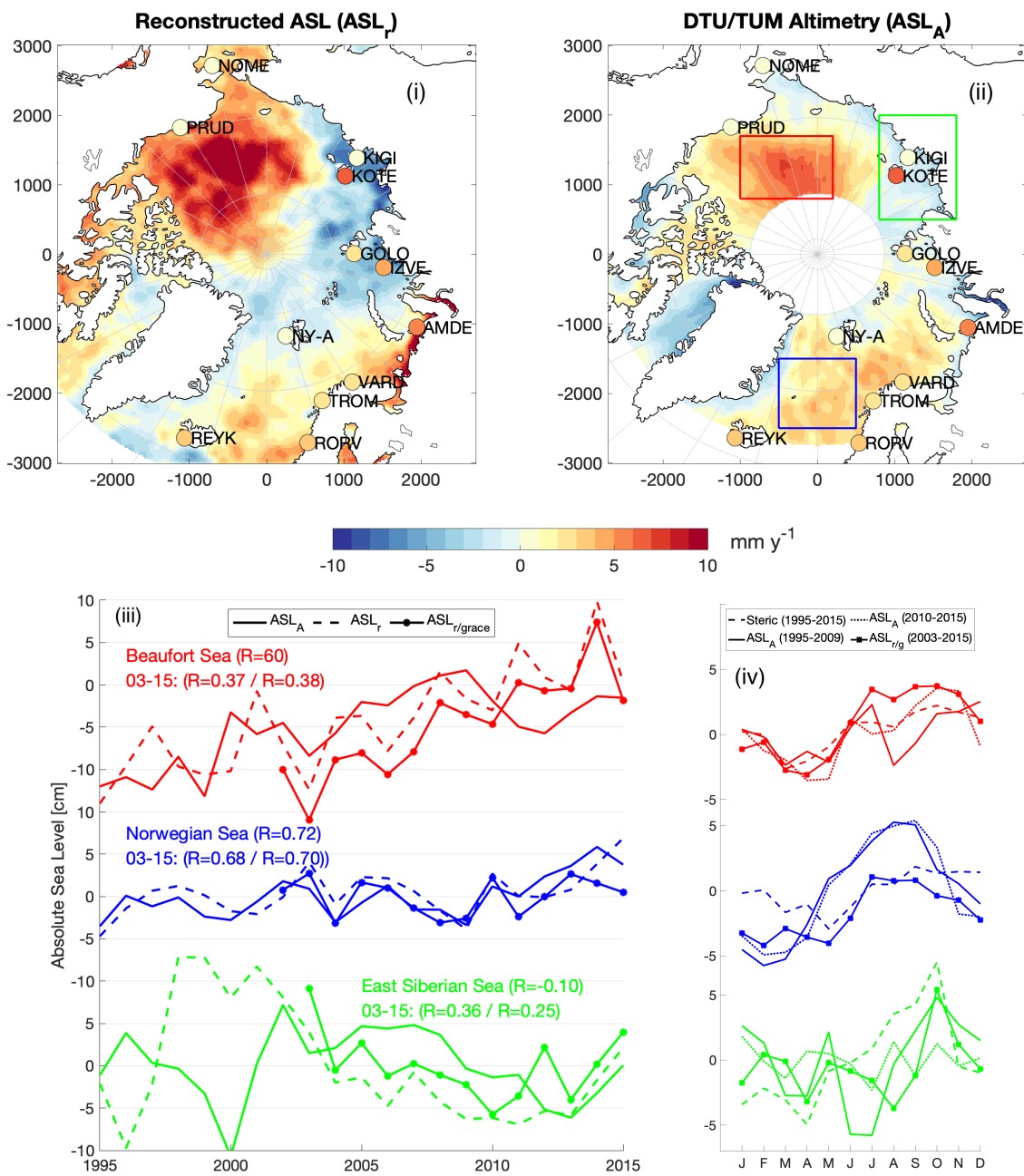

**Figure 7.** Absolute sea level trend of the reconstructed product ($\dot{ASL}_r$) (first map from left (i)) and from DTU/TUM Altimetry ($\dot{ASL}_A$) (second map (ii)) from 1995 to 2015 [mm y$^{-1}$]. In both maps is the sea level trend of the 12 VLM-corrected tide gauges ($\dot{ASL}_{TG}$) shown with circles. Third panel from left (iii) shows the timeseries of $ASL_A$, $ASL_r$ and $ASL_{r/grace}$ (ASL reconstructed with GRACE (mean of the two GRACE estimates used in this study) for three selected regions, Beaufort Sea (red), Norwegian Sea (blue) and East Siberian Sea (green) (areas marked in the DTU/TUM Altimetry map). The top R-coefficient for each region shows the correlation between $ASL_A$ and $ASL_r$ and beneath is shown the R-coefficient between $ASL_A$ and $ASL_r$ / $ASL_A$ and $ASL_{r/grace}$ for 2003 to 2015. The right panel (iv) shows the mean seasonal cycle for two periods of $ASL_A$ (solid line: 1995-2009, dotted line: 2010-2015), $ASL_{r/grace}$ ($ASL_{r/g}$) and steric sea level (Steric) for the same three regions as in (iii).

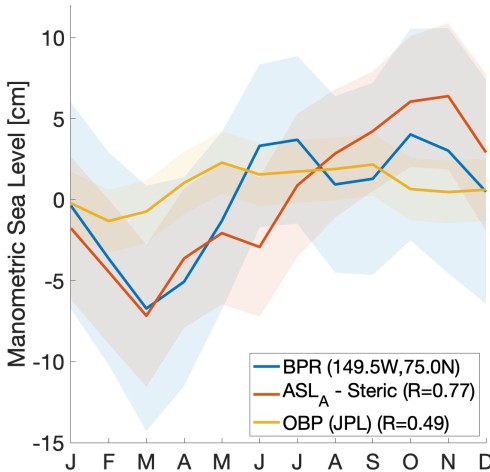

**Figure 8.** Mean seasonal cycle of manometric sea level from Beaufort Sea bottom pressure recorder from the Beaufort Gyre Exploration Project, Mooring A (149.5W,75.0N) (blue), DTU Altimetry (ASL$_A$) - Steric (red), GRACE JPL Mascon RL06 (yellow). Shaded areas indicate one standard deviation. Values from altimetry, steric and GRACE are averages from 50 km around the BPR mooring location. Correlations (R) with BPR are given for GRACE and altimetry minus steric.

Bottom pressure recorders (BPRs) deployed by Woods Hole Oceanographic Institution in the Beaufort Sea used to validate GRACE in the Arctic by (Peralta-Ferriz and Morison, 2010; Peralta-Ferriz et al., 2014) provides an independent estimate of the manometric sea level change. Because of sensor drift and small changes in location, the BPRs are not usable for detection of trends over longer time periods (Proshutinsky et al., 2019) and therefore not comparable with the manometric sea level
reconstruction from figure 5. Instead, the mean seasonal cycle of BPR is compared to manometric sea level from GRACE and altimetry (ASL$_A$) minus steric in the Beaufort Sea (figure 8). It shows the bottom pressure variations of the BPR correlates better with an altimetry minus steric estimate (inverted from eq. 2) (R=0.77) compared to the GRACE-estimate of seasonal manometric sea level change (R=0.49).

### 4.1    Comparing observed and reconstructed manometric sea level change

The reconstructed manometric sea level trend ($\dot{M}$, figure 5f) is varying between 0 and 2 mm y$^{-1}$, with small spatial variability. The reconstructed manometric contribution is generally much smaller than the estimates from GSFC mascons (RL05) (Luthcke et al., 2013) used by Raj et al. (2020) and CSR RL05 (Save et al., 2016) preferred in Carret et al. (2017). The two RL06 solutions shown in figure 6, are more consistent than the RL05 solutions shown in Ludwigsen and Andersen (2020), but still show significant differences. The trends disagree in particular along the Eastern Arctic coastlines and the Beaufort Gyre. This
is also where the largest residuals between the reconstructed SLT and altimetry is observed (figure 9), hence is no obvious manometric SLT derived from GRACE that is able to explain the residuals between $\dot{ASL}_A$ and $\dot{ASL}_r$.

The annual manometric sea level change of two GRACE solutions and the reconstructed estimate for three selected regions in the Arctic Ocean (shown in the bottom right panel of figure 6) show temporal similarities in the Beaufort and Norwegian Sea, while a large manometric sea level peak in both GRACE estimates in 2012 in the East Siberian Sea is not explained by the manometric reconstruction. Also opposite manometric sea level signals are evident in particular in 2005-2006 in all regions. The annual cycle of the reconstructed manometric sea level is dominated by the inverse barometer effect, while the fluctuations in geoid (N) are an order of magnitude smaller (top-right panel of figure 6) than the total reconstructed manometric signal. However, the trend estimate ($\dot{N}$) is significantly larger than the IB-trend at most locations (figure 5).

Figure 5g shows that ECCO has larger manometric sea level rise in the interior of the Arctic Ocean, while the coastal zones, except East Siberia, are slightly lower than $\dot{M}$. The ECCO-model includes a dynamic sea level change associated with wind-forcing and ocean currents (Forget et al., 2015). Those dynamic changes are not part of $\dot{M}$ and is probably the main reason for the difference between ECCO OPB and $\dot{M}$ seen in figure 5h. The dynamic mass variations follows largely the temporal variations of the AO (Peralta-Ferriz et al., 2014; Armitage et al., 2018). To some extent, the coastal/non-coastal Arctic dipole from Peralta-Ferriz et al. (2014) is recognized in figure 5h, but over the extent of the time series of this study, the effect of the AO is assumed to be less significant than the pattern in Peralta-Ferriz et al. (2014), which relies on only 7 years of data. A dynamic mass contribution will also be more significant in the trend of the GRACE-estimates, that only spans 13 years. A significant manometric sea level rise in the interior of the Arctic Ocean, is also recognized by the GRACE-estimates in figure 6. This is also where GRACE has reached the best correlation with in-situ bottom pressure recorders (Peralta-Ferriz et al., 2014), probably because (false) land-leakage corrections will be less relevant in the interior of the Arctic Ocean. Since the altimetry record is limited to 81.5°N, there is no validation of the performance of the reconstructed SLT estimate north of 81.5°N, but from figure 9 we see that the reconstructed SLT is lower than altimetry in most of the visible interior of the Arctic Ocean. The high latitudes of the Beaufort Sea is the exception, where the steric contribution in itself is enough to explain the altimetry observed SLT and adding a significant manometric sea level rise would increases the residual to altimetry.

Figure 5a-c shows that the contributions from contemporary ice loading has a (compared to steric) small contribution to spatial sea level variability, but the sea level fingerprints (Mitrovica et al., 2011) from deglaciation of Greenland and glaciers are, however, still clearly visible with a sea level fall of 0.5 to 1 mm y$^{-1}$. This seems to be qualitatively in agreement with regional sea level fingerprint studies of Bamber and Riva (2010); Spada (2017); Frederikse et al. (2017). In total, the three figures sums to a sea level rise of around 1 mm y$^{-1}$ in most of the Arctic, except in areas close to land-deglaciation (like Greenland and Svalbard). From the top right panel of figure 6, is is seen that the sea level contribution from ice changes is accelerating and is almost 0.2 mm y$^{-1}$ by the end of this period. From the comparison with GRACE (figure 6) we see that GRACE has a more significant sea level fall in coastal regions with land-deglaciation. It is likely that the GRACE-estimates are affected by leakage from land mass that is falsely interpreted as an manometric sea level change.

## 4.2 Comparing reconstructed absolute sea level with altimetry

For 87% of the area of the Arctic between 65N° and 82N° is the reconstructed sea level pattern ($\dot{ASL}_r$) in agreement with the observed sea level trend ($\dot{ASL}_A$) within the 68% confidence interval (figure 9). The main difference between $\dot{ASL}_r$ and $\dot{ASL}_A$

is the mentioned larger sea level rise (residual of + 5-10 mm y$^{-1}$) in the Beaufort Sea and sea level fall (residual of - 2-5 mm y$^{-1}$) in the East-Siberian seas of $\dot{\text{ASL}}_r$. In the Norwegian Sea the residuals are in the order of +/- 1.5 mm y$^{-1}$, which because of the low uncertainty in the area falls outside the 68% confidence interval in large areas.

The spatial correlation coefficient (R) between $\dot{\text{ASL}}_r$ and $\dot{\text{ASL}}_A$ is 0.50 (R=0.23 without the halosteric contribution) and R=0.53 when using the ECCO OBP estimate instead of the reconstructed manometric sea level from 1995-2015. The correlation coefficient falls to R=0.38 when limiting the period from 2003-2015. The correlation coefficients reached by Ludwigsen and Andersen (2020) using different release-05 GRACE-mascons from 2003-2015 (R=0.19-0.40) combined with the same steric and a altimetric dataset. When the SLT is reconstructed with release-06 GRACE mascons ($\dot{\text{ASL}}_{r/grace}$) the correlation is R=0.37 for GSFC and R=0.34 for JPL and thereby slightly lower than with the reconstructed manometric contribution. This reflects the fact that trend estimates are naturally more sensitive over shorter timeseries, and in particular when the sea level is as dynamic as in the Arctic Ocean.

Before the era of SAR altimetry (pre CryoSat-2, launched in October 2010), the ability to separate the leads and the sea ice was more difficult due to the larger footprint of the conventional satellites. Therefore, in areas with a dense sea ice cover (like the Beaufort Sea), more altimetric observations exist during the sea level high of the autumn and fewer during winter/spring where sea level is lower (e.g. Armitage et al. (2016)). The sampling of the seasonal signal (figure 7 panel (iv)) can create a seasonal bias which was more pronounced before the CryoSat-2 era, because of the lower resolution in the pre-SAR era. This bias can contribute to a flattening of the trend in the Beaufort Sea as seen from the time series in figure 7 panel (iii). In figure 7 panel (i) and (ii) $\dot{\text{ASL}}_A$ shows a lower trend in the Beaufort Sea than $\dot{\text{ASL}}_r$, mainly caused by an apparent sea level decline from 2010-2015. Studies of altimetry-based sea level in the Beaufort Sea from Giles et al. (2012) and Armitage et al. (2016) indicate a similar flattening of the sea level anomaly around 2009/10. The change in sea level trend is attributed to a shift in the cyclonic regime of the Beaufort Gyre in 2010/2011 (Proshutinsky et al., 2015) which released significant amounts of freshwater (Armitage et al., 2016). However, the significant change in the Beaufort Sea coincides with the transition from Envisat to CryoSat-2 and a inter-satellite bias in DTU/TUM Altimetry can not be excluded, and thereby contributing to the poor correlation after 2009.

Ludwigsen and Andersen (2020) showed a better agreement with the SLT of Armitage et al. (2016) in the Beaufort Sea than the DTU/TUM estimate in the present study, using the same steric product and different GRACE-estimates. The residuals between $\dot{\text{ASL}}_r$ and $\dot{\text{ASL}}_A$ (of this study) are however seemingly smaller than the results from Raj et al. (2020) who found region-averaged residuals in the Beaufort Sea of +10 mm y$^{-1}$ from 2003-2009 and +3.6 mm y$^{-1}$ from 2010-2016 between a GSFC GRACE mascons together with a steric model-estimate and the same DTU/TUM altimetry product. Carret et al. (2017) also found that the sea level change from altimetry had less spatial variability than the combination of mass+steric from both 1992-2014 and 2003-2010.

The mean seasonal cycle of the Beaufort Sea (panel (iv) of figure 7) shows how a summer and wintertime peak of $\text{ASL}_A$ (in January and June) is visible before 2010, but almost disappears in the CryoSat-2 era. The double-peak is also found by Armitage et al. (2016) from 2003 to 2014, but is not nearly as large because of the relative larger CryoSat-2 weight. Since the manometric components is yearly averaged, only the seasonal variations of the steric component of $\text{ASL}_r$ is shown. From the

figure, it is evident that the steric signal is dominating the seasonal variation in the Beaufort Sea. A significant residual between steric sea level and $ASL_A$ indicates a dominant manometric signal i the North Atlantic, in alignment of the results of (Carret et al., 2017), who found that the variability in the North Atlantic (GNB-sector) is predominantly non-steric.

$A\dot{S}L_A$ (figure 7) shows a sea level rise in the Norwegian Sea that extends until it reaches the sea ice boundary, which
(intentionally) coincides with the average SAR-boundary of CryoSat-2. From altimetry it is unclear if this signal is a real physical signal or due to a bias when different altimetric observations (different satellites and SAR/conventional), sea ice and open ocean regions are aligned (no sea state bias correction in the SAR areas) in the DTU/TUM altimetry product or a known error in the SAR-based DTU18MSS (Andersen et al., 2018) that is used as a reference in the altimetry data. $A\dot{S}L_r$ shows a similar SLT-pattern in the Norwegian Sea from a combination of the thermosteric change (warmer ocean) (figure 4) and
a sea level fall from a gravitational weakening of Greenland (figure 5a). The boundary between sea ice and open ocean is however less significant in $A\dot{S}L_r$ and a spatial bias in altimetry can therefore not be excluded. A thermosteric sea level rise that is countered by a halosteric sea level fall in the Norwegian Sea is also reported by the other studies (Henry et al., 2012; Carret et al., 2017; Raj et al., 2020). The residuals in the present study are however qualitatively smaller than the results of the mentioned studies, albeit they use different subsets of periods and for the case of Raj et al. (2020) only basin-wide averages
are given.

## 4.3 Comparing ASL-trends in coastal regions

TGs are only able to observe coastal sea level change, which is often disturbed by the local environment that might be unknown (e.g. small river outflow, local construction, packing of sea ice etc.), which affects both sea level measurements from TGs and altimetry.
In figure 10 and table 1, the contributions to $A\dot{S}L_r$ are quantified at the location at each of the 12 TGs by taking the mean trend of a radius of 50 km (5 km for GIA and elastic VLM). This radius ensures, that Rorvik, Nome and Reykjavik overlaps the altimetric data, but the fewer number of data points might cause the altimetry estimates at these TGs to be more variable. The residuals between the TG-observed ASL-trend, $A\dot{S}L_{TG}$, and $A\dot{S}L_r$ are visible from figure 7. $A\dot{S}L_{TG}$ is in agreement of $A\dot{S}L_r$ at only 5 of the 12 TGs (8 of 12 for $A\dot{S}L_A$ / $A\dot{S}L_{TG}$) are within the combined standard error, while 9 are within two standard
errors (95 % confidence interval). Relative low standard errors of $A\dot{S}L_{TG}$ contributes to the apparent low agreement.

The Norwegian tide gauges (Rorvik, Tromso, Vardo, Ny Ålesund) are together with Reykjavik the most consistent with the smallest errors. These are also the sites where $ASL_A$ and $ASL_r$ are most precise, due to little or no sea ice and high density of hydrographical data (figure 3). For Rorvik and Vardo, is $A\dot{S}L_r$ more in alignment with $A\dot{S}L_{TG}$ than $A\dot{S}L_A$, while $A\dot{S}L_{TG}$ of Tromso and Ny Ålesund is better aligned with $A\dot{S}L_A$. We see that for Vardo and Rorvik, the $A\dot{S}L_r$ is split between a steric and
355 a mass contribution of roughly the same size, which is similar to the contributions share of the global sea level trend (Church and White, 2011b; WCRP, 2018). At Tromso a local negative halosteric trend (more saline water) is lowering $A\dot{S}L_r$, while for the area around Tromso (50-200 km), $A\dot{S}L_r$ agrees well with the observed $A\dot{S}L_{TG}$ and $A\dot{S}L_A$.

The Siberian coast has multiple river outlets that contributes with significant freshwater of the Arctic Ocean (Proshutinsky et al., 2004; Morison et al., 2012; Armitage et al., 2016). A positive halosteric sea level trend is visible at the coast of the

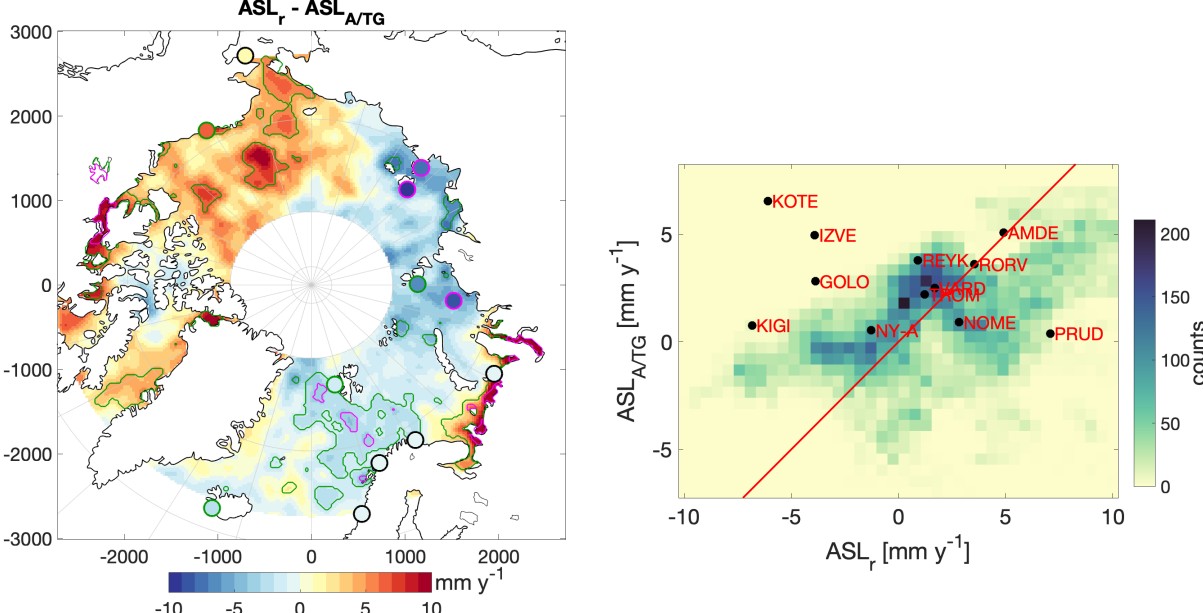

**Figure 9.** Left map shows the difference between $\dot{ASL}_r$ and $\dot{ASL}_{A/TG}$. The green contour shows the areas or tide gauges (green edge) where the absolute difference is larger than one standard error (68% confidence interval), but less than two standard errors (95% confidence interval) (combined error from figure 11). The magenta areas or tide gauges (magenta edge) where the absolute difference is larger than two standard errors. Right panel shows a correlation matrix between $\dot{ASL}_r$ and $\dot{ASL}_{A/TG}$. The color indicates the number of data grid cells falling into bin size of 0.5 mm y$^{-1}$. 96% of the grid cells with data is covered within the bounds of the matrix ($N_{total}$=18150). The red line is where $\dot{ASL}_r$ is equal to $\dot{ASL}_{A/TG}$.

Bering and Kara Sea (figure 4), where the river OB has a major outflow. At Amderma TG, which is located on the coast between the Barents and Kara Sea, an apparent large halosteric sea level fall is also recognized by the TG-measured sea level, despite rather large errorbars due to lack of in situ data (figure 4). Iceloss from Novaya Zemlya contributes with over 1 gigaton of freshwater to the Kara Sea every year and the ice loss has been accelerating (Melkonian et al., 2016), but the contribution is small compared to the +500 Gt coming from the rivers every year. The halosteric signal could (falsely) be extrapolated from the

gulf of Ob which has major river outlets and the agreement with $\dot{ASL}_{TG}$ is accidental. The halosteric sea level rise at Anderma remains doubtful, since $\dot{ASL}_A$ shows a negative ASL-trend in opposition to $\dot{ASL}_{TG}$ and $\dot{ASL}_r$.

    The four TGs along the eastern Siberian coast (Izvestia Tsik, Golomianyi, Kotelnyi, Kigiliah) all observe a rising sea levels, while $\dot{ASL}_A$ and in particular $\dot{ASL}_r$ shows a negative trend in the region. Missing data in the end of the timeseries of Golomianyi (figure 2) might significantly alter the observed trend. From 2005-2010, Golomanyi showed a sea level fall, while few high

measurements in 2012 and 2014 skews the trend upwards. Also the TG Izvestika Tsik observed a decreasing sea level from 2006/7-2013, but an apparent steep sea level increase from 2013-2015 changes the trend to positive.

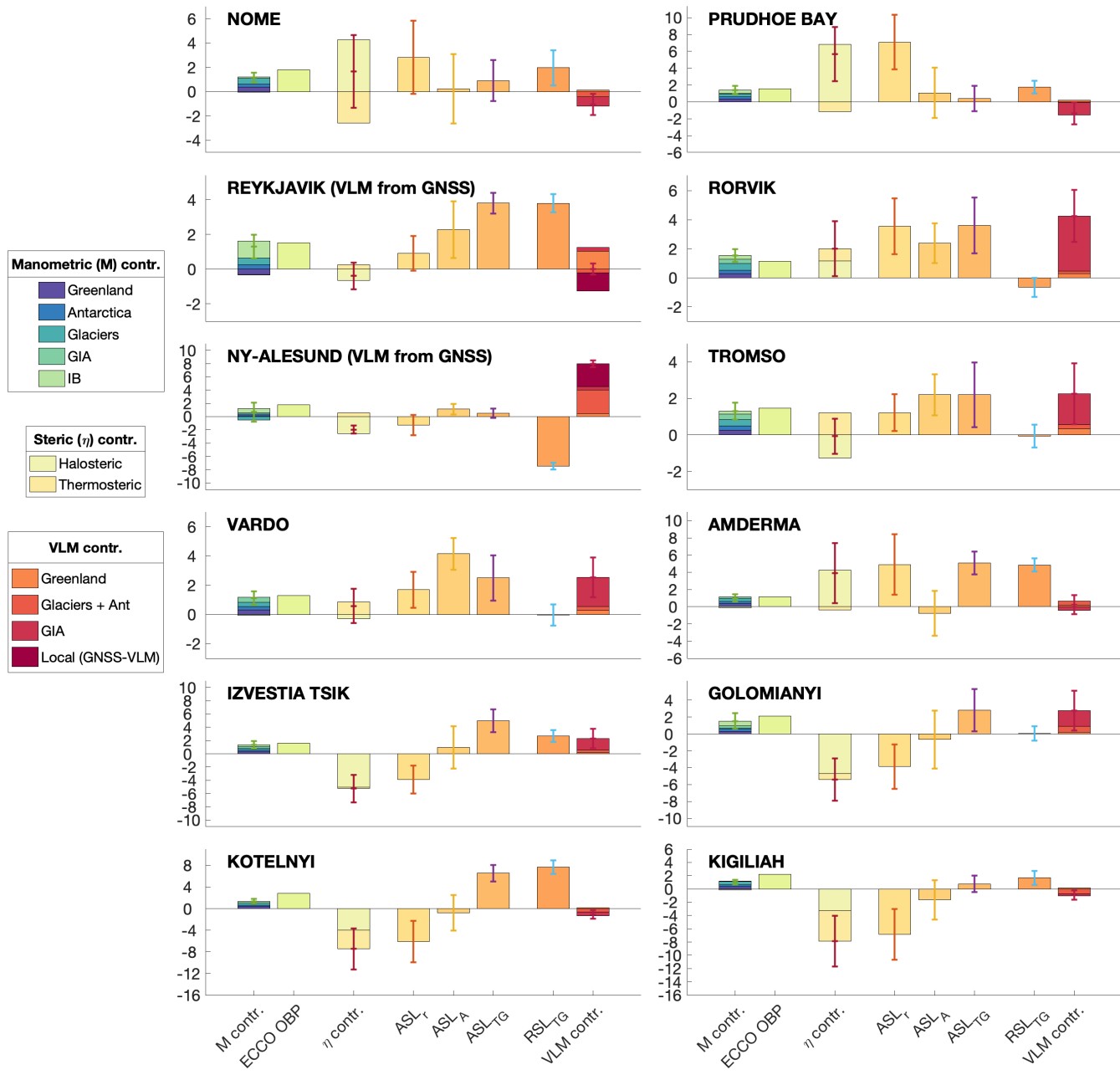

**Figure 10.** Components of sea level trend [mm y$^{-1}$] for each tide gauge from 1995-2015. The three bars in the middle (A$\dot{S}$L$_r$, A$\dot{S}$L$_A$ and A$\dot{S}$L$_{TG}$) are the three independent estimates of absolute sea level. The error bars indicate one standard error (combined error from each component when relevant) equivalent to the 68% confidence level. The VLM component 'Local (GNSS-VLM)' is only relevant at Reykjavik and Ny Ålesund, because significant local properties causes VLM that is not present in the VLM-model (Ludwigsen et al., 2020b). Glacier component of VLM includes the effect of rotational feedback, ocean loading, and Antarctica which is less than 0.5 mm y$^{-1}$ combined.

| | $\dot{\text{RSL}}_{TG}$ | VLM (model/GNSS) | $\dot{\text{ASL}}_{TG}$ | IB | $\dot{N}$ | $\dot{M}$ | $\dot{\eta}$ | $\dot{\text{ASL}}_r$ | $\dot{\text{ASL}}_A$ |
|---|---|---|---|---|---|---|---|---|---|
| NOME | $2.0 \pm 1.4$ | $-1.1 \pm 0.9$ | $\mathbf{0.9 \pm 1.7}$ | 0.1 | $1.1 \pm 0.4$ | $1.2 \pm 0.4$ | $1.7 \pm 3.0$ | $\mathbf{2.8 \pm 3.0}$ | $\mathbf{0.2 \pm 2.8}$ |
| PRUDHOE BAY | $1.7 \pm 0.8$ | $-1.4 \pm 1.3$ | $\mathbf{0.4 \pm 1.5}$ | 0.4 | $1.0 \pm 0.5$ | $1.4 \pm 0.5$ | $5.7 \pm 3.2$ | $\mathbf{7.1 \pm 3.2}$ | $\mathbf{1.1 \pm 3.0}$ |
| REYKJAVIK | $3.8 \pm 0.5$ | $0.0 \pm 0.3$ | $\mathbf{3.8 \pm 0.6}$ | 1.0 | $0.3 \pm 0.7$ | $1.3 \pm 0.7$ | $-0.4 \pm 0.8$ | $\mathbf{0.9 \pm 1.0}$ | $\mathbf{2.3 \pm 1.6}$ |
| RORVIK | $-0.7 \pm 0.7$ | $4.3 \pm 1.8$ | $\mathbf{3.6 \pm 1.9}$ | 0.3 | $1.3 \pm 0.4$ | $1.5 \pm 0.4$ | $2.0 \pm 1.9$ | $\mathbf{3.5 \pm 1.9}$ | $\mathbf{2.4 \pm 1.4}$ |
| NY-ALESUND | $-7.4 \pm 0.5$ | $8.0 \pm 0.5$ | $\mathbf{0.5 \pm 0.7}$ | 0.6 | $0.1 \pm 1.4$ | $0.7 \pm 1.4$ | $-2.0 \pm 0.6$ | $\mathbf{-1.3 \pm 1.5}$ | $\mathbf{1.1 \pm 0.8}$ |
| TROMSO | $-0.1 \pm 0.6$ | $2.3 \pm 1.7$ | $\mathbf{2.2 \pm 1.8}$ | 0.1 | $1.1 \pm 0.5$ | $1.3 \pm 0.5$ | $-0.1 \pm 1.0$ | $\mathbf{1.2 \pm 1.0}$ | $\mathbf{2.2 \pm 1.1}$ |
| VARDO | $-0.0 \pm 0.7$ | $2.5 \pm 1.4$ | $\mathbf{2.5 \pm 1.5}$ | -0.1 | $1.2 \pm 0.5$ | $1.1 \pm 0.5$ | $0.6 \pm 1.2$ | $\mathbf{1.7 \pm 1.2}$ | $\mathbf{4.1 \pm 1.1}$ |
| AMDERMA | $4.9 \pm 0.8$ | $0.2 \pm 1.1$ | $\mathbf{5.1 \pm 1.3}$ | -0.1 | $1.1 \pm 0.4$ | $1.0 \pm 0.4$ | $3.9 \pm 3.5$ | $\mathbf{4.9 \pm 3.5}$ | $\mathbf{-0.8 \pm 2.6}$ |
| IZVESTIA TSIK | $2.7 \pm 0.9$ | $2.3 \pm 1.5$ | $\mathbf{5.0 \pm 1.7}$ | 0.2 | $1.1 \pm 0.6$ | $1.3 \pm 0.6$ | $-5.2 \pm 2.1$ | $\mathbf{-3.9 \pm 2.1}$ | $\mathbf{1.0 \pm 3.2}$ |
| GOLOMIANYI | $0.0 \pm 0.9$ | $2.8 \pm 2.3$ | $\mathbf{2.8 \pm 2.5}$ | 0.6 | $0.9 \pm 0.9$ | $1.5 \pm 0.9$ | $-5.4 \pm 2.5$ | $\mathbf{-3.9 \pm 2.6}$ | $\mathbf{-0.7 \pm 3.4}$ |
| KOTELNYI | $7.7 \pm 1.3$ | $-1.1 \pm 0.8$ | $\mathbf{6.5 \pm 1.5}$ | 0.2 | $1.1 \pm 0.4$ | $1.4 \pm 0.4$ | $-7.5 \pm 3.8$ | $\mathbf{-6.1 \pm 3.8}$ | $\mathbf{-0.8 \pm 3.3}$ |
| KIGILIAH | $1.7 \pm 1.0$ | $-0.9 \pm 0.7$ | $\mathbf{0.8 \pm 1.3}$ | -0.1 | $1.2 \pm 0.4$ | $1.0 \pm 0.4$ | $-7.9 \pm 3.8$ | $\mathbf{-6.8 \pm 3.8}$ | $\mathbf{-1.6 \pm 3.0}$ |

**Table 1.** 1995-2015 sea level trends [mm y$^{-1}$] at the 12 tide gauge locations. The trends (least-squares) are generally based on a annual mean-value of a 50 km radius around the tide gauge. For VLM a 5 km radius is used, except for Ny-Alesund and Reykjavik where VLM is based on GNSS-measurements. The columns in bold indicate the three estimates of Absolute sea level ($\dot{\text{ASL}}_{TG}$, $\dot{\text{ASL}}_r$ and $\dot{\text{ASL}}_A$). Errors indicate the 1 standard error equivalent to the 68% confidence level.

Non-seasonal variations in sea level in eastern Siberian seas are dominated by large scale wind patterns controlled by the AO (Volkov and Landerer, 2013; Peralta-Ferriz et al., 2014; Armitage et al., 2018). These wind-driven sea level effects are largely manometric, but are not included in the manometric estimate ($\dot{M}$). Wind-driven sea level change is part of ECCO OBP, which is 1-2 mm y$^{-1}$ higher than $\dot{M}$ in the area (figure 5), while GRACE-trends from 2003 to 2015 are ranging from -2 (JPL) to +2 (GSFC) mm y$^{-1}$ and might also be affected by leakage. $\dot{\text{ASL}}_{TG}$ is however around 5 mm y$^{-1}$ lower than $\dot{\text{ASL}}_r$ in the East Siberian Arctic and therefore not explained by the reconstructed manometric sea level difference to GRACE/ECCO.

The positive ASL trend among tide gauges in the eastern Russian Arctic is consistent with the results of other studies using an extended set of Russian tide gauges (Proshutinsky et al., 2004; Henry et al., 2012). Remarkably, the TG-trend at Kotelnyi and Kigiliah differ with almost 6 mm y$^{-1}$ (in total 12 cm difference over the time span of this study) despite being less than 250 km apart. From the timeseries in figure 2 a 30 cm RSL rise from 2002 to 2008 at Kotelnyi is visible. This significant change is however not observed by any altimeter product. A reasonable explanation can be local coastal subsidence caused by thawing of permafrost or oil depletion, which also could explain the mentioned sea level 'jumps' of Golomanyi and Izvestika Tsik. This is however speculative, since it is not confirmed by any literature. In general, the poorest agreement is found at the Siberian TGs, which is similar to Armitage et al. (2016) who found that these tide gauge correlated the least with the altimetric observations. The sea level drop in year 2000 observed by Altimetry in East Siberian Sea (figure 7 panel (iii)) is also to some extent seen by Kotelnyi and Kigiliah TGs (figure 2) that are located in the same region, which indicates that poor T/S measurements in the region has lead to a an false steric sea level high in the region from 1998-2002.

Nome and Prudhoe Bay in Alaska both show a positive steric TG-trend which is not reflected in $\dot{\mathrm{ASL}}_{\mathrm{TG}}$ or $\dot{\mathrm{ASL}}_{\mathrm{A}}$, thus resulting in a rather large discrepancy between $\dot{\mathrm{ASL}}_{\mathrm{r}}$ and $\dot{\mathrm{ASL}}_{\mathrm{A/TG}}$. The strong halosteric trend of the Beaufort Gyre, might be extrapolated towards the Alaskan coastline and into the Bering Strait in the DTU steric model. There is no evidence in the literature for an extent of the Beaufort Gyre doming as shown from the halosteric trend, which indicates, that the weighted spatial interpolation in combination with higher hydrographic data density in the Beaufort Sea creates this widening of the Beaufort Gyre.

Ny-Ålesund on Svalbard is dominated by a large VLM caused by recent deglaciation. This uplift completely mitigates the large sea level fall measured by the tide gauge and results in small rise of $\dot{\mathrm{ASL}}_{\mathrm{TG}}$. In (Ludwigsen et al., 2020a) it is argued that the discrepancy between GNSS and the VLM-model in large originates from VLM because of post-LIA deglaciation on Svalbard (Rajner, 2018). This viscoelastic GIA-like LIA-effect will certainly also have a gravitational sea level fingerprint ($\dot{N}$) that should be added to the manometric SLTs $\dot{\mathrm{M}}$. This can explain some of the difference between $\dot{\mathrm{ASL}}_{\mathrm{r}}$ and $\dot{\mathrm{ASL}}_{\mathrm{A/TG}}$. A possibly positive dynamic $\dot{\mathrm{M}}$-change (from the (ECCO OBP)$-\dot{\mathrm{M}}$ difference in figure 5h) could further close the gap between $\dot{\mathrm{ASL}}_{\mathrm{r}}$ and $\dot{\mathrm{ASL}}_{\mathrm{A/TG}}$.

None of the TG-sites in this study experience a net sea level fall due to contemporary deglaciation and GIA ($\dot{N}$ in table 1) and only Ny-Ålesund (-0.4 mm y$^{-1}$) and Reykjavik (-0.2 mm y$^{-1}$) will experience a small sea level fall from contemporary deglaciation alone. So even though the Arctic is heavily prone to ice mass loss and thus a weakened gravitational pull, the Arctic as a region is not experiencing an absolute sea level fall from comtemporary deglaciation. On the contrary, it causes the sea level to rise with around 1 mm y$^{-1}$ in most of the Arctic. However, by accounting for the deglaciation effect on VLM, contemporary deglaciation will contribute to an relative sea level fall in most areas of the Arctic.

## 5 Uncertainty of the contributions

The uncertainties of the trend estimates for RSL$_{\mathrm{TG}}$, VLM, gravitational fingerprint ($N$), steric ($\eta$) in table 1 and figure 10 are derived as the standard error of the detrended and deseasoned timeseries of the contributions. GIA (Caron et al., 2018) and altimetry (Rose et al., 2019) has a associated uncertainty that is used. For the VLM-model a 10% error is added to account for uncertainties of the earth model (Wang et al., 2012).

The spatial distribution of the uncertainties are shown in figure 11. Generally, the largest uncertainties are found along the Siberian coast, with the steric uncertainty in most cases being the largest source of uncertainty (figure 10). The standard error naturally reflects if the steric heights are unstable and poorly constrained (if for example there are few hydro-graphic data (figure 3)). In principle, this method requires temporal independence, which is not entirely true, since outliers are replaced with data from adjacent years. Furthermore, large influence by the non-periodic and non-linear Arctic Oscillation, would enhance the uncertainty, even though this is a real physical signal. Thereby is the estimated error a composite of uncertainties originating from the way the sea level component is constructed and from interannual variability.

The mass contribution and VLM has naturally the largest uncertainties close to glaciated areas. Glacial ice loss on Baffin Island is poorly constrained in the ice model, which is reflected with large uncertainties in this area. The uncertainty of altimetry

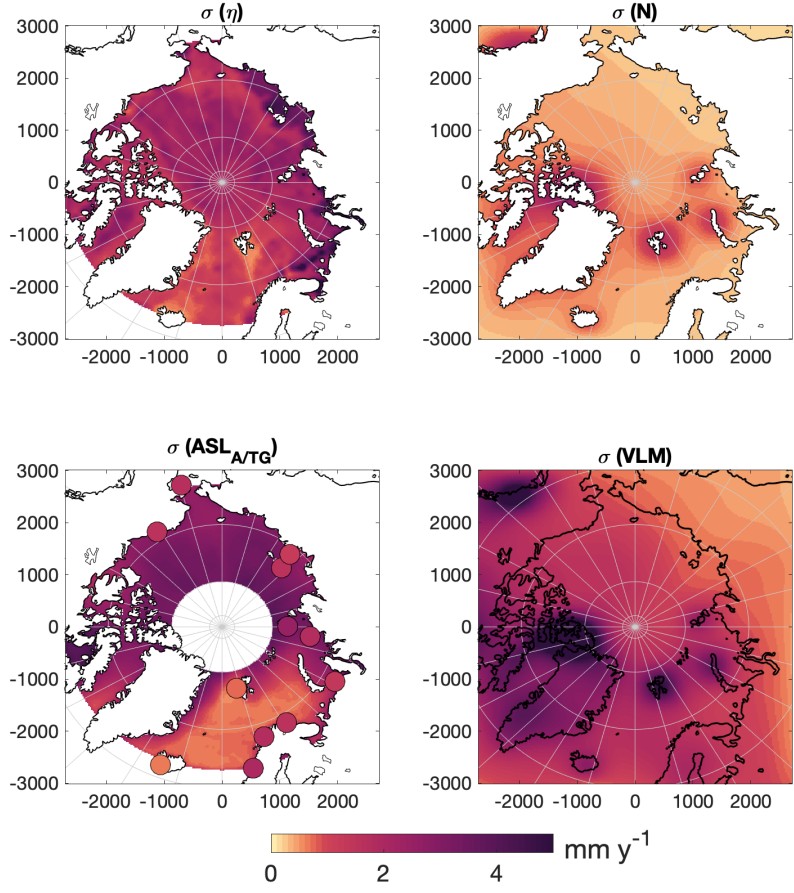

**Figure 11.** Standard error (68% confidence interval) of the 1995-2015 trend [mm y$^{-1}$] for combined steric, combined $\dot{N}$, A$\dot{S}$L$_{A/TG}$ and combined VLM contributions.

is reflecting the data availability of areas with sea ice contrary the ice-free ocean, while the largest uncertainties of the TGs are those with largest interannual variability.

## 6  Conclusion

All significant contributions to the sea level change from 1995-2015 in the Arctic Ocean were mapped and assessed at 12 tide gauges located throughout the Arctic Ocean. As the first study, the observed sea level was attempted reconstructed without the use of GRACE data or modeled steric data in a region where observations are sparse and very uncertain. Thus is it possible to attribute the observed sea level changes to their origin and understand the components of the altimetry and TG-observed sea level trend.

Figure 7 shows that the spatial pattern of altimetry observed sea level trend ($A\dot{S}L_A$) is generally restored from the reconstructed trend-estimate ($A\dot{S}L_r$). The spatial correlation between the reconstructed trend map and altimetry derived trend (R=0.50) from 1995-2015 is outperforming a similar analysis for GRACE-based reconstructions from 2003-2015 (R=0.34-0.37), however, when using the reconstructed manometric sea level component instead of GRACE, a similar correlation is reached (R=0.38). Hence is the calculated manometric contribution an equal alternative to GRACE that should be considered for studying long-term past and future Arctic sea level change.

Figure 9 shows the residual between observed sea level ($A\dot{S}L_{A/TG}$) and the reconstructed ASL estimate within the combined uncertainty. The reconstructed ASL-trend agrees with altimetry at 87% of the area within the 68% confidence level (96% of the area within the 95% confidence level). The residual map indicate an improvement over previous studies (Carret et al., 2017; Raj et al., 2020), however this assessment is only qualitatively since different subsets of periods are used. The two major residuals between altimetry and the reconstructed product are found in the Beaufort Sea and East Siberian Sea. In both regions, the altimetry estimate by Armitage et al. (2016) has a better agreement than the used DTU/TUM-altimetry product. A dominant halosteric trend in the Beaufort Sea, that is larger than the altimetric trend is also observed by Carret et al. (2017) and Raj et al. (2020).

The sea level trend at 5 (9 using the 95% confidence interval) of the 12 VLM-corrected TGs agree with the reconstruction, while 8 of 12 TGs agree with altimetry. The relatively poor correlation at TG sites, can be attributed sparse T/S-data to compute steric sea level along the coast of the Beaufort Sea and the Siberian Arctic and possible local unknown land subsidience/uplift affecting the tide gauge record.

From figure 10 and figure 11 it is evident that the steric estimate is the main source of uncertainty. Some areas, in particular, the Norwegian Sea, has more observations (from both altimetry and hydrographic data) and thus are the individual contributions estimated with lower uncertainty. The Siberian Seas are however poorly constrained with observations and both the steric product, altimetry and tide gauges show large uncertainties. The manometric sea level change has a more uniform and smaller contribution to ASL with smaller associated uncertainties compared to the steric component. However, considering the difference to GRACE-estimates and modelled ECCO-estimate, is the uncertainty of the total manometric contribution also significant and its reasons are not yet resolved. Except for the central Arctic Ocean, where GRACE is less affected by leakage-related issues, is GRACE not able to explain the obtained residuals.

Generally, would the Arctic sea level reconstruction be improved, if steric estimate is further constrained, since it is the dominant feature of Arctic Ocean sea level change. Eventually integrating sea surface temperature and salinity from satellite observations could improve the estimates in areas with few in-situ data. Furthermore an independent estimate of the dynamic contribution to manometric sea level change is needed to include the significant wind-driven sea level changes in the Arctic.

*Code and data availability.* Tide gauge sea level timeseries is available at psmsl.org, the VLM-model available at data.dtu.dk/articles/dataset/ Arctic_Vertical_Land_Motion_5x5_km, DTU Steric is available at ftp.space.dtu.dk/pub/DTU19/STERIC/, the REAR-software is available at github at github.com/danielemelini/rear.git.

*Author contributions.* CBL: Method, concept, data analysis and writing. OBA: Concept and editing. SKR: Providing altimetry data, validation and editing.

*Competing interests.* The authors declare no competing interests.

*Acknowledgements.* The authors want to thank three anonymous reviewers for their comments that greatly improved the manuscript. Furthermore, we thank Lambert Caron for providing the Caron2018 GIA-model, available at https://vesl.jpl.nasa.gov/solid-earth/gia/ and Danielle Melini for creating the REAR-code (Melini et al., 2015). This research was funded by the EU-INTAROS project (Grant agreement no. 727890) (CAL and OBA) and by the ESA-Climate Change Initiative sea level budget closure (Expro RFP/3-14679/16/INB) (OBA and
470 SKR).

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
