# Peer review of "Components of 21 years (1995-2015) of Absolute Sea Level Trends in the Arctic"

_Ocean Science, 2020_

## Referee Comment (RC1) · Anonymous Referee #1 · 15 Dec 2020

This study uses tide gauges from 12 Arctic coastal locations along with an altimetry-based absolute sea level product to infer the contributions to absolute sea level trends over 1995-2015 at those sites. The novelty of the study comes from using an approach to determine the mass component of the sea level without using GRACE, meaning that the timeseries can be extended and is not reliant on selecting one of the GRACE products, which can vary greatly. After introducing the various datasets and initial breakdown into steric and mass components, the authors further compute the steric (halo- and thermo-), and mass (sources from different deglaciation sites, atmospheric loading, and dynamic mass contribution) components at each location, and discuss possible reasons for the trends there.

This paper presents an interesting approach to investigate the varying causes of sea

level trends at each of the Arctic tide gauge sites in the study. However, I feel that the manuscript is unclear in quite a number of places, and the main results section (section 6) reads more as a list of what happens at each location, with little effort to point to the main notable findings and what we should infer from them. Therefore, by the end of the conclusion I was not completely sure what the main new results were, other than that uncertainty is large. I do think there are some interesting results in this manuscript but, as it stands, they are somewhat buried in the text. Below I have listed some key things which I feel need to be addressed to make the paper more accessible to the reader. Note that there are also some grammar issues which should be fixed during the review.

Main comments

1) The structure of the paper needs to be clearer - it is hard to know where the sections are divided, and whether we are reading about background, methods or results. In particular, the introduction seems to be made up of two introduction-style paragraphs with the rest as methods. I would suggest re-ordering so that you introduce the ASL formula before you discuss the mass component, and adding some more motivation about why it is important to study this before having a sentence or two to explain the layout of the rest of the paper. This would help to clarify what the reader can expect, and may result in less confusion. Introducing each subsequent section/subsection by saying what will be discussed will also help. Section 2 is very brief and I feel needs to describe in more detail how the data is obtained, especially in ice-covered areas. Perhaps combining sections 2 and 3 into 'methods' or 'data' would help with the weighting of methods versus results.

2) As part of my issue with clarity, it seems that section 6 is a large amount of information with no real synthesis apart from what appears in the conclusion. At the very least, a summary such as that in the conclusion should appear before the end of section 6, as otherwise the manuscript jumps from a pure description of the figures to the conclusion without all of the individual results being brought together in a coherent way. What actually are the main results of the analysis? How do they fit into the current literature? I also feel the last two paragraphs of section 6 - a very welcome discussion on uncertainties - should be given their own subsection and figure 8 referred to before the conclusion (which should not be introducing new results!). In this part, it is important to discuss Table 1 and the large error bars at some of the sites - how much of this error is due to uncertainties in the methods compared to drawing a linear trend over strong inter annual variability (which is evident in some timeseries in Figure 2)? If they cannot easily be separated quantitatively, this should at least be stated.

3) I feel as though the reader is left to infer for themselves what some of the figures show. In particular, Figure 4 is clearly important, as it shows the contribution from each of the mass components (the notable difference between this study and those that use GRACE-based products), but it is never given any real explanation or interpretation in the text. Despite this, results from two of the sub-panels are referred to in the abstract (if my understanding is correct), which is bad practice at the very least. Each figure should have at least a few lines dedicated to what it is showing scientifically, and should be referred to adequately when it is being described. For example, lines 152-153 discuss a positive halosteric trend along Siberia – I would have assumed that this is referring to Figure 3, but in that figure much of the Siberian coast has a negative trend. Unless it is referring to a particular station in Figure 6? Being specific about which part of each figure is being referred to would greatly help the reader in section 6 where multiple figures from multiple datasets seem to be being discussed.

4) The naming of each component/combination of components needs to be more consistent. For example, line 146 mentions 'derived product' but this seems to be used interchangeably between the one that uses mass from ECCO and mass from the REAR model. In the conclusion it is then described as 'derived steric estimate and in the mass product'. Similarly, for figure 6, the caption describes steric+geoid+dyn, but on the figure it's steric+geoid+dym [sic] + IB. As there are various different acronyms and sums of components throughout the study, explicitly naming these and then referring to them by one term only from then on would aid keeping track of what is being described.

5) With regards to Figure 5, I think more effort should be made to demonstrate that the cause of the discrepancy is the hypothesised overestimation of ASL in the altimetry data pre-2011. Can you somehow compare the final years where Cryosat-2 is available to see if the ASL (if not the trend itself) is comparable? What are the implications of the discrepancy on the later analysis? This should be stated in the text.

6) Figure 8 is not well explained, either in what the subplots show or what they mean for the comparison of the different products. It should appear before the conclusion, and the spatial patterns described explicitly, at the very least where there are notable differences. Lines 200-203 state "Without the use of ECCO, the derived product agrees with altimetry at 98% of the area, while only 5 out of 12 of the TG-data agree with derived product. For the steric+mass(ECCO) product, the products agree at 99% of the area and at 11 out of 12 TG's." Why do we see no visible difference in the steric+mass versus steric+mass(ECCO) in Figure 5, but a big difference in agreement with tide gauges by this calculation? The interpretation of this figure and its consequence should be made more explicit.

Line by line comments

Abstract: lines 6-7: This was never explicitly stated in the paper Lines 8-9: Again, this is never stated in the paper, and not discussed. I believe it is being taken from the bottom right panel of figure 4? This should at least be stated somewhere in the main text. Last line: What is the conclusion/importance of this work?

Line 45: IB has not yet been introduced

Lines 46-47: Referencing of figure 5 is out of order, which may be against journal rules

Line 50: GNSS has not been explicitly introduced as an acronym

Line 60: How many months are missing? Could you provide some idea of this and how it may affect the results? Is the missing data predominantly in one season?

Lines 61-64: Be clear that you are talking about relative sea level here and not absolute.

Is there a reason you choose to discuss this rather than the absolute sea level?

Lines 121-125: This description of the middle panel of figure 5 and its caption are not clear; which terms are used? Is it just the ECCO Ocean Bottom Pressure (bottom left of figure 4) that is added to the steric data, as may be implied by line 125 and the caption? Or is the dynamic mass contribution from ECCO included in the figure, as line 122 might suggest? Referencing the relevant panel on figure 4 would help to clarify this.

Line 129: Dooming -> doming

Lines 195-197: These figures show that the steric uncertainty can reach 10 mm/yr, which is the maximum halosteric contribution to the trend in Figure 3 (and will exceed the sum of the two steric components in places). How much impact does this have on the interpretation of the results?

Figure 1b: I would suggest displaying the same latitude limit on this figure as the other maps in the paper. Having the more southerly latitudes does not add anything to the story as far as I can tell, and zooming in on the northerly latitudes would make the squared markers easier to see

Figure 2: It would be easier to read this figure if the y axes were all the same. Why are the trend lines centred on 0? It was initially confusing to see some trends (for example, at Ny-Alesund) being offset from the data. Some of the trends are very small and having the offset makes it hard to see. I see the error in each is noted in Table 1, but why do the yellow lines on figure 2 have a different trend to the corresponding column in table 6?

Figure 8: It is hard to distinguish between dark blue and black on lower panels - it may be clearer to change them to a different colour

---

## Referee Comment (RC2) · Anonymous Referee #2 · 17 Dec 2020

I had hoped that this manuscript would prove interesting from reading the Abstract, but I regret to say that I was disappointed. The authors promised to extend the record of sea level change in the Arctic, while decomposing that change into mass and steric components. However, there are numerous problems with the manuscript. The material is disordered; many acronyms are used before they are defined, or not defined at all; ECCO is used without any calibration or validation, assuming it to be somehow "correct", or least better than GRACE; many maps are shown, but no time series; no material comparisons are made with other publications in terms of results, to show whether or how the new material is an improvement over existing results; references and relevant comparisons are missing, such as the work by authors like Laxon and Giles to address between-mission differences in altimetric height measurements; prior

results describing Arctic mass changes from GRACE are not discussed (work by Morison); there are incorrect statements like "mass change (also called Ocean Bottom Pressure" and "ASL measured by satellite altimetry is measured relative to the Earth's center".

Again with regret - this work reads like a technical report that has not properly matured into a scientific study, and I do not think that it merits publication.

––––––––––––––––––––––––––––––

---

## Author Comment (AC1) · 26 Feb 2021

This study uses tide gauges from 12 Arctic coastal locations along with an altimetry-based absolute sea level product to infer the contributions to absolute sea level trends over 1995-2015 at those sites. The novelty of the study comes from using an approach to determine the mass component of the sea level without using GRACE, meaning that the timeseries can be extended and is not reliant on selecting one of the GRACE products, which can vary greatly. After introducing the various datasets and initial breakdown into steric and mass components, the authors further compute the steric (halo- and thermo-), and mass (sources from different deglaciation sites, atmospheric loading, and dynamic mass contribution) components at each location, and discuss possible reasons for the trends there.

This paper presents an interesting approach to investigate the varying causes of sea level trends at each of the Arctic tide gauge sites in the study. However, I feel that the manuscript is unclear in quite a number of places, and the main results section (section 6) reads more as a list of what happens at each location, with little effort to point to the main notable findings and what we should infer from them. Therefore, by the end of the conclusion I was not completely sure what the main new results were, other than that uncertainty is large. I do think there are some interesting results in this manuscript but, as it stands, they are somewhat buried in the text. Below I have listed some key things which I feel need to be addressed to make the paper more accessible to the reader. Note that there are also some grammar issues which should be fixed during the review.

The authors thanks reviewer #1 for the thorough and constructive comment on the manuscript. It is clear that the manuscript needed substantial edits and that in particular the structure of the paper was unclear. We hope the reviewer will find that the edits has improved the paper significantly and made it more accessible and interesting. A pdf with all changes highlighted is attached as supplement and the specific comments are addressed below:

Main comments 1) The structure of the paper needs to be clearer - it is hard to know where the sections are divided, and whether we are reading about background, methods or results. In particular, the introduction seems to be made up of two introduction-style paragraphs with the rest as methods. I would suggest re-ordering so that you introduce the ASL formula before you discuss the mass component, and adding some more motivation about why it is important to study this before having a sentence or two to explain the layout of the rest of the paper. This would help to clarify what the reader can expect, and may result in less confusion. Introducing each subsequent section/subsection by saying what will be discussed will also help.

The introduction is completely altered and the method and data section is separated in a more coherent way.

2a) As part of my issue with clarity, it seems that section 6 is a large amount of information with no real synthesis apart from what appears in the conclusion. At the very least, a summary such as that in the conclusion should appear before the end of section 6, as otherwise the manuscript jumps from a pure description of the figures to the conclusion without all of the individual results being brought together in a coherent way. What actually are the main results of the analysis? How do they fit into the current literature? I also feel the last two paragraphs of section 6 - a very welcome discussion on uncertainties - should be given their own subsection and figure 8 referred to before the conclusion (which should not be introducing new results!).

The results section (now section 4) is divided into a subsection comparing the reconstructed sea level trend with altimetry and that compares ASL-trends at TG-locations. Section 5 contains an assessment of the observed and reconstructed ASL-trends with respect to the observed uncertainties (figure 7 and 8).

References to existing literature has been added when relevant – in particular where the observations disagree with the reconstructed sea level.

2b) In this part, it is important to discuss Table 1 and the large error bars at some of the sites - how much of this error is due to uncertainties in the methods compared to drawing a linear trend over strong inter annual variability (which is evident in some timeseries in Figure 2)? If they cannot easily be separated quantitatively, this should at least be stated.

We have added a paragraph on this (line 280-282). Furthermore a computational error was found when calculating the errors of TG, which made them too low.

3a) I feel as though the reader is left to infer for themselves what some of the figures show. In particular, Figure 4 is clearly important, as it shows the contribution from each of the mass components (the notable difference between this study and those that use GRACE-based products), but it is never given any real explanation or interpretation in the text. Despite this, results from two of the sub-panels are referred to in the abstract

(if my understanding is correct), which is bad practice at the very least.

In the results section several paragraphs describing figure 4 has now been added.

3b) Each figure should have at least a few lines dedicated to what it is showing scientifically, and should be referred to adequately when it is being described. For example, lines 152-153 discuss a positive halosteric trend along Siberia – I would have assumed that this is referring to Figure 3, but in that figure much of the Siberian coast has a negative trend. Unless it is referring to a particular station in Figure 6? Being specific about which part of each figure is being referred to would greatly help the reader in section 6 where multiple figures from multiple datasets seem to be being discussed.

In this particular case, we discuss a a positive halosteric trend in western Russian Arctic between Kara and Barents Sea (Amderma TG). This has now been specified (line 231-235). Generally it is specified when describing changes at specific locations.

4) The naming of each component/combination of components needs to be more consistent. For example, line 146 mentions 'derived product' but this seems to be used interchangeably between the one that uses mass from ECCO and mass from the REAR model. In the conclusion it is then described as 'derived steric estimate and in the mass product'. Similarly, for figure 6, the caption describes steric+geoid+dyn, but on the figure it's steric+geoid+dym [sic] + IB. As there are various different acronyms and sums of components throughout the study, explicitly naming these and then referring to them by one term only from then on would aid keeping track of what is being described.

Beginning with the method section and throughout the manuscript, the naming has been completely changed is hopefully now more intuitive to follow. The term 'manometric sea level' is introduced and we use 'reconstructed sea level' instead of 'derived'. ECCO OBP has been removed from the results and most of the analysis since it looked like ECCO was the 'true' sea level. We define the manometric (incl. IB) +steric as the reconstructed ASL (ASLr) which is compared to ASL from altimetry (ASL_A) and tide-gauges (ASL_TG).

5) With regards to Figure 5, I think more effort should be made to demonstrate that the cause of the discrepancy is the hypothesised overestimation of ASL in the altimetry data pre-2011. Can you somehow compare the final years where Cryosat-2 is available to see if the ASL (if not the trend itself) is comparable? What are the implications of the discrepancy on the later analysis? This should be stated in the text.

We have added a timeseries of the reconstructed ASL and altimetric ASL in figure 5 for two regions which identifies the drop in altimetry over the Beaufort Gyre in 2010.

6) Figure 8 is not well explained, either in what the subplots show or what they mean for the comparison of the different products. It should appear before the conclusion, and the spatial patterns described explicitly, at the very least where there are notable differences. Lines 200-203 state "Without the use of ECCO, the derived product agrees with altimetry at 98% of the area, while only 5 out of 12 of the TG-data agree with derived product. For the steric+mass(ECCO) product, the products agree at 99% of the area and at 11 out of 12 TG's." Why do we see no visible difference in the steric+mass versus steric+mass(ECCO) in Figure 5, but a big difference in agreement with tide gauges by this calculation? The interpretation of this figure and its consequence should be made more explicit.

The difference between ECCO/non-ECCO was small on +/- 10 mm/y scale shown in figure 8. As said, is the ECCO estimate removed from most of the analysis and also from figure 5 and 8.

Line by line comments Abstract: lines 6-7: This was never explicitly stated in the paper Lines 8-9: Again, this is never stated in the paper, and not discussed. I believe it is being taken from the bottom right panel of figure 4? This should at least be stated somewhere in the main text. Last line: What is the conclusion/importance of this work?

The abstract has been completely changed and the conclusions are now specifically stated in the last paragraph.

Line 45: IB has not yet been introduced.

Ok.

Lines 46-47: Referencing of figure 5 is out of order, which may be against journal rules.

This is referenced to avoid double-showing the altimetric trend. The text has been altered and a parenthesis is added around the reference. Hope this is ok.

Line 50: GNSS has not been explicitly introduced as an acronym

Ok.

Line 60: How many months are missing? Could you provide some idea of this and how it may affect the results? Is the missing data predominantly in one season?

Missing months are now indicated in figure 2 and mentioned in text (line 113-114).

Lines 61-64: Be clear that you are talking about relative sea level here and not absolute. Is there a reason you choose to discuss this rather than the absolute sea level?

Text has been improved, so that it is more clear that RSL is discussed and how it changes when VLM is added.

Lines 121-125: This description of the middle panel of figure 5 and its caption are not clear; which terms are used? Is it just the ECCO Ocean Bottom Pressure (bottom left of figure 4) that is added to the steric data, as may be implied by line 125 and the caption? Or is the dynamic mass contribution from ECCO included in the figure, as line 122 might suggest? Referencing the relevant panel on figure 4 would help to clarify this.

This has been changed according to the other associated changes made.

Line 129: Dooming -> doming Corrected

Lines 195-197: These figures show that the steric uncertainty can reach 10 mm/yr, which is the maximum halosteric contribution to the trend in Figure 3 (and will exceed

the sum of the two steric components in places). How much impact does this have on the interpretation of the results?

More discussion on uncertainty has been added.

Figure 1b: I would suggest displaying the same latitude limit on this figure as the other maps in the paper. Having the more southerly latitudes does not add anything to the story as far as I can tell, and zooming in on the northerly latitudes would make the squared markers easier to see

You are right. We have changed it to display the same area as the other maps.

Figure 2: It would be easier to read this figure if the y axes were all the same. Why are the trend lines centred on 0? It was initially confusing to see some trends (for example, at Ny-Alesund) being offset from the data. Some of the trends are very small and having the offset makes it hard to see. I see the error in each is noted in Table 1, but why do the yellow lines on figure 2 have a different trend to the corresponding column in table 6?

Figure 2 had several computational issues in the first place. These have alle been fixed, and figure 2, figure 6 and table 1 should all agree now.

Figure 8: It is hard to distinguish between dark blue and black on lower panels - it may be clearer to change them to a different colour

Different colorscale is selected for figure 8.

Please also note the supplement to this comment:
https://os.copernicus.org/preprints/os-2020-87/os-2020-87-AC1-supplement.pdf

---

## Author Comment (AC2) · 26 Feb 2021

I had hoped that this manuscript would prove interesting from reading the Abstract, but I regret to say that I was disappointed. The authors promised to extend the record of sea level change in the Arctic, while decomposing that change into mass and steric components. However, there are numerous problems with the manuscript. The material is disordered; many acronyms are used before they are defined, or not defined at all; ECCO is used without any calibration or validation, assuming it to be somehow "correct", or least better than GRACE; many maps are shown, but no time series; no material comparisons are made with other publications in terms of results, to show whether or how the new material is an improvement over existing results; references and relevant comparisons are missing, such as the work by authors like Laxon and

[Figure]

Giles to address between-mission differences in altimetric height measurements; prior results describing Arctic mass changes from GRACE are not discussed (work by Morison); there are incorrect statements like "mass change (also called Ocean Bottom Pressure" and "ASL measured by satellite altimetry is measured relative to the Earth's center". Again with regret - this work reads like a technical report that has not properly matured into a scientific study, and I do not think that it merits publication.

The manuscript has been completely altered and the more specific points in the comment by reviewer #2 have all been addressed. Since the above comment has a more general character, we don't list all the changes made, but refer to points addressed by reviewer #1 and the attached pdf with track changes. We hope that the reviewer will find the manuscript significantly improved.

Please also note the supplement to this comment:
https://os.copernicus.org/preprints/os-2020-87/os-2020-87-AC2-supplement.pdf

**Supplement:**

[revised manuscript text omitted]

$$\dot{M} = \dot{N}_{\mathrm{NHG}} + \dot{N}_{\mathrm{GRE}} + \dot{N}_{\mathrm{SH}} + \dot{N}_{\mathrm{GIA}} + \mathrm{I}\dot{B} \qquad (12)$$

240   Figure 4g shows the OBP-trend from the ECCOv4r4-model (Estimating the Circulation and Climate of the Ocean (ECCO) version 4 release 4) (Forget et al., 2015; Fukumori et al., 2019), which is used to estimate the dynamic contribution to sea level . The dynamic mass change is mainly a wind-driven effect that significantly changes the spatial distribution of ocean mass (Calafat et al., 2012; Dangendorf et al., 2014; Armitage et al., 2018) - also on secular time scales. a model estimate of $\dot{M}$. The difference between ECCO and $\dot{M}$ is displayed in 4h.

245   Because the ECCO-model is among other forced by wind (Forget et al., 2015), we use the difference between ECCO and

**4   Results**

Generally, the steric (in particular the halosteric) sea level trend is dominating the spatial variability of the sum of $\dot{N}$ and IB as an estimate of the dynamic contribution to mass ($D\dot{M}$, bottom right map of figure 4 reconstructed sea level trend ($\mathrm{ASL}_r$), with over 10 mm y$^{-1}$ in the Beaufort Gyre and -7 mm y$^{-1}$ in the Russian Arctic (figure 3).

250   ## 5   Comparison of estimates of the Arctic Absolute Sea Level Trend

Two derived trend estimates of the ASL budget is created from steric + mass ($\dot{\eta} + \dot{N} + I\dot{B}$) (without the dynamic component) and steric + mass (ECCO), where ECCO is used as the mass component and hence includes dynamic mass changes In contrast, is the reconstructed manometric sea level trend ($\dot{M}$) varying between 0 and 2 mm y$^{-1}$, with smaller spatial variability. This is in alignment with the 2003-2015 OBP-estimates from GRACE JPL mascons (Wiese et al., 2016) used in Ludwigsen and Andersen (2020)
255   , but way smaller than the estimates from GSFC mascons (Luthcke et al., 2013) used by Raj et al. (2020) and CSR (Save et al., 2016) used by Carret et al. (2017).

Figure 4a-c shows that the contributions from contemporary ice loading has a (compared to steric) small contribution to spatial sea level variability, but the sea level fingerprints from deglaciation of Greenland and glaciers are however til clearly visible with a absolute sea level fall of 0.5 to 1 mm y$^{-1}$, which seems to be in agreement with global sea level fingerprint
260   studies of Bamber and Riva (2010); Spada (2017); Frederikse et al. (2018). In total, the three figures sums to a sea level rise of around 1 mm y$^{-1}$ in most of the Arctic, except close to areas with deglaciation (like Greenland and Svalbard). They are compared to the independent estimates of ASL change from TG corrected for VLM and altimetry

Figure 4g shows that ECCO has a higher manometric sea level change in the interior of the Arctic Ocean, while the coastal zones, except east Siberia, are lower than $\dot{M}$.

265   The two derived ASL-trend estimates are shown ECCO-model does attempt to include term dynamic sea level changes associated with wind-forcing and ocean currents into their OBP-estimate (Forget et al., 2015). Those changes are not part of $\dot{M}$ and is probably the main reason for the difference between ECCO OPB and $\dot{M}$.

[Figure]

**Figure 5.** Absolute sea level trend  the reconstructed product (AṠL$_r$) (left) and from  DTU/TUM Altimetry (AṠL$_
[revised manuscript text omitted]

---

## Referee Report (RR1)

**Report\_Ludwigsen**

This revised version has neen significantly improved compared to the original manuscript. However it still needs substantial improvement before being considered as publishable. My main comments concern : (1) the writing, sometimes poor and unclear, with several vague or not argued statements (see below), (2) the lack of discussion on the data uncertainties, in particular on the steric component, and (3) the lack of acknowledgement to similar previously published studies on arctic sea level (just quoted in the reference list but without discussion/comparison) (see below).

- The abstract is vague and non informative. Nowhere in the abstract is explained what the computed reconstructed sea level consists of. The abstract says : 'as the first study...', a wrong statement in view of the many previously published studies. In addition, it claims that NOT using GRACE is a progress, but why ? GRACE has a nearly global coverage over the Arctic and provides unique information about tha mass component of the sea level budget in the Actic. Why oppose GRACE to the approach considred here ? The only acceptable argument is the shorter time series of GRACE data. I recommend to rewrite the abstract and clearly explain the approach considered in this study and presensing the main outcomes (and eventually the novel results compared to previous studies). More detailed comments below :
- In the introduction, describe the main objective of this study and the approach used
- Line 31: what is the meaning of 'difficult' in the sentence 'Previous attemps in reconstructing sea level in the Arctic have show to be difficult' ? Explain. In the same sentence, the author write : ... because satellite and in situ observations are less consistent'. Less consistent than what ? Curious claim considering that the authors ALSO use satellite and in situ observations...
- Line 35 : Clarify what you want to say by : 'the 10 mm/yr discrepancies between GRACE solutions'. In absolute value, this has no meaning. It should be compared to the amplitude of the signal. It also depends where.
- Line 37 : I disagree with the sentence ' some authors tend to choose the GRACE solution that closes the sea level budget. This not true. Please remove this sentence.
- -Method section, lines 63-64 and Equation 7 : What about local VLMs , unrelated to GIA and ongoing mass redistribution ? These could be important at some TGs
- Eqution 6 : What about the GIA component for altimetry (the component of -0.3 mm/yr removed to altimetry data in terms of global mean)? It seems to be ignored here.
- Line 90. Fig. 1 should be quoted first.
- Section 3.1, Altimetry. What is the accuracy of the altimetry data set in this region partly covered by ice. The authors disgard GRACE because of its 'poor' accuracy, but what about altimetry ?
- Section 3.1, VLMs. The authors choose the option of using theroretical VLMs due to GIA and ongoing mass redistributions rather than GNSS data. Ok but explain why using the model is best and at least compare where possible the model VLMs with GNSS.
- 3.3, steric sea level. The interesting part of this study is the use of situ T/S profiles over the Arctic. However, we have no idea of the data coverage nor on the accuracy of this data set. What is the integration depth H ? I strongly recommend the authors provide information on the T/S data set, a crucial aspect of the study.
- Section 3.4, ocean mass component. This section is hard to follow, with several unclear sentences, e.g., lines 163-165.

- Results section. I strongly recommend to put into perspective the results of this study with the published literature (e.g., Henry et al, Armitage et al., Carret et al., Raj et al). What is novel ? Wht is different ? etc.

---

## Author Response (AR2)

**Author response in red**

The authors thanks the referee for the thorough review and positive assessment. Parts of the manuscript has been altered completely, which makes some of the comments below irrelevant for the updated manuscript. All comments below have either been addressed and the manuscript subsequently changed, or the comments are no longer relevant due to substantial manuscript changes.

The authors have made substantial changes to the structure of the manuscript and some figures and it is now significantly clearer to read and understand. In particular, the motivation in the introduction is more coherent and I appreciate the care taken to clarify the naming of different terms. In general, my previous comments have been addressed and I think this is an interesting study worthy of publication. I have a few remaining questions/comments but these are generally very minor and I have outlined them by line number below.

Note that there are very minor typos in the manuscript (generally related to plurals) that I have not listed but should be picked up during the final proofing.

Line-by-line comments:

Line 5: "that is independent from any observed sea level change". This is a little confusing, since the reconstruction is based on observations. What is independent from what? By the observed sea level change here, do you mean tide gauges? Some clarification would be welcome

Line 73: Why is GRACE listed as a satellite product that is used in the study?

Line 78: should there be a dot above ASL_A? In general, I recommend checking the variables from equations that appear in the main text for this. Also check equations 8 and 9

Line 130: I may be wrong, but I think the Armitage and Davidson (2014) reference here, and all subsequent references to that paper, should actually be referencing the Armitage et al (2016) paper in the reference list

Line 163-165: these sentences are confusing. Please reword, and refer to the subpanel of figure 4 that the terms discussed are related to

Line 173: this is the first time ECCO has been introduced. A bit more information about this product would be welcome, especially as further on the discrepancy between ECCO and the manually-calculated Mdot are discussed

Line 177: Figure 5 should be referred to here in relation to ASL_r

Line 179: Refer to figure 4f here

Line 192: has anyone else found such a difference before? Are there any references for the sort of magnitude difference you could expect from comparing two such datasets?

Line 207: this dataset is from 2003-2014. Here it should be explained to the reader why the altimetry-based dataset of Armitage et al (2016) does not suffer from the seasonal bias and 'flattening' you describe.

Lines 199-207: This paragraph focuses on key differences in the Beaufort Gyre region, but there is no mention of the paper by Giles et al (2012) who specifically focus on trends from satellites in the region. This paper should be discussed here.

Line 217: specify the interior Arctic Ocean, or it suggests TGs have no use in the study

Lines 225-226: which error estimate are you talking about here? You state the tide gauges, but it is ASL_A that has the lowest error estimates for the specified locations, not ASL_TG…

Line 233: please reference figure 3 here

Line 242: I am surprised that this section does not refer to Figure 2. For these four stations in particular, the timeseries in Figure 2 is variable and one could argue that at Izvestia Tsik leaving out the last year or two of the timeseries would significantly alter the linear trend,

while Golomianyi has a lot of missing data towards the end which may skew it upwards. Some acknowledgement of this figure would be welcome here

Line 242-244: it may be worth reminding the reader why the manometric estimate from ECCO and the summed estimate could differ here

Line 294: there have been studies of

Line 297: what was the time period used for the previous correlations? Could that have affected the results? This should be stated here as it is important for the conclusion

Conclusion: Given it is the main motivation for the study, I feel that a sentence or two in the conclusion could be dedicated to emphasising the novelty of this study and how it does/does not improve on the GRACE-based results in the literature. At the moment, a sentence on an improved correlation is given but is then followed by a discussion of uncertainties, which dampens the message. Perhaps moving the first two sentences of the second paragraph of the conclusion to the end of the first would help

Figure 4: the caption says "pherical glaciers" which I think should be "peripheral glaciers"? Also, I am a bit confused by how c) and d) look so different but have a very similar yearly contribution

Figure 5: is there a reason that the colormap has a different scale to figure 3? Providing both figures on the same scale would make the relative contribution of halosteric sea level change easier to see

Technical corrections
Line 20: "sea ice floats" -> "sea ice floes"?

Line 112-113: "and therefore does not significantly affect the trend estimates with a seasonal bias" -> "and therefore the trend estimates are not significantly affected by a seasonal bias"

Line 125: "does all TGs show" -> "all TGs show"

Line 129: "300.000" -> "300,000"

Line 130: "spatial and temporal" -> "spatially and temporally"

Line 134: "which is cause to large uncertainties" -> "which results in large uncertainties"

Line 150: brackets are unclear. Do you mean "…changes in ice loading (equation 4, similar to the elastic VLM-component) is computed…"?

Line 181: "way smaller" -> "much smaller"

Line 184: "til" -> "still"; line 185: "a absolute" -> "an absolute"

Line 202: "i.e." -> "e.g."

Line 203: erroneous bracket to be removed

Line 239: "Left map" -> "The left map"

Line 319: "thus can the individual contributions" -> "thus the individual contributions can"

**Response to Interactive comment on* "Assessment of 21 years of Arctic Ocean Absolute Sea Level Trends (1995–2015)" *by* Carsten Ankjær Ludwigsen et al.**

**Anonymous Referee #3**
**Author response in red**

The authors thanks reviewer #3 for the comments on the manuscript, which lead to substantial edits of the manuscript.

This revised version has neen significantly improved compared to the original manuscript. However it still needs substantial improvement before being considered as publishable. My main comments concern : (1) the writing, sometimes poor and unclear, with several vague or not argued statements (see below), (2) the lack of discussion on the data uncertainties, in particular on the steric component, and (3) the lack of acknowledgement to similar previously published studies on arctic sea level (just quoted in the reference list but without discussion/comparison) (see below).

- The abstract is vague and non informative. Nowhere in the abstract is explained what the computed reconstructed sea level consists of. The abstract says : 'as the first study...', a wrong statement in view of the many previously published studies. In addition, it claims that NOT using GRACE is a progress, but why ? GRACE has a nearly global coverage over the Arctic and provides unique information about tha mass component of the sea level budget in the Actic. Why oppose GRACE to the approach considred here ? The only acceptable argument is the shorter time series of GRACE data. I recommend to rewrite the abstract and clearly explain the approach considered in this study and presensing the main outcomes (and eventually the novel results compared to previous studies). More detailed comments below :

The abstract has been altered. The main reason for not using GRACE is the disagreement between available GRACE-products. This has been highlighted throughout the manuscript.

- Line 31 : what is the meaning of 'difficult' in the sentence 'Previous attemps in reconstructing sea level in the Arctic have show to be difficult' ? Explain. In the same

sentence, the author write : ... because satellite and in situ observations are less consistent'. Less consisent than what ? Curious claim considering that the authors ALSO use satellite and in situ observations...

- Line 35 : Clarify what you want to say by : 'the 10 mm/yr discrepancies between GRACE solutions'. In absolute value, this has no meaning. It should be compared to the amplitude of the signal. It also depends where.
- Line 37 : I disagree with the sentence ' some authors tend to choose the GRACE solution that closes the sea level budget. This not true. Please remove this sentence.

Introduction has been rewritten which should solve the issues raised in the three comments above.

- -Method section, lines 63-64 and Equation 7 : What about local VLMs , unrelated to GIA and ongoing mass redistribution ? These could be important at some TGs
A paragraph on this has been added, despite little information on VLMs from non-glacial change.

- Eqution 6 : What about the GIA component for altimetry (the component of -0.3 mm/yr removed to altimetry data in terms of global mean)? It seems to be ignored here.
Not entirely sure what is meant here, but the altimetric product is without any GIA-correction.

- Line 90. Fig. 1 should be quoted first.
OK

- Section 3.1, Altimetry. What is the accuracy of the altimetry data set in this region partly covered by ice. The authors disgard GRACE because of its 'poor' accuracy, but what about altimetry ?
It has been highlighted that altimetry is challenged over sea ice and that the associated uncertainties are larger. Also disagreements with Armitage et al 2016 has been described in the results section.

- Section 3.1, VLMs. The authors choose the option of using theroretical VLMs due to GIA and ongoing mass redistributions rather than GNSS data. Ok but explain why using the model is best and at least compare where possible the model VLMs with GNSS.
For detailed VLM/GNSS comparisons in the Arctic from 2003-2015 the reader is referred to Ludwigsen et al, 2020 (GRL). GNSS-data does often not extend to prior 2002/3, which is why the VLM-model is used, even in regions like the Norwegian Coast where GNSS is located close to the TG. Ny-Ålesund and Reykjavik are exceptions.

- 3.3, steric sea level. The interesting part of this study is the use of situ T/S profiles over the Arctic. However, we have no idea of the data coverage nor on the accuracy of this data set. What is the integration depth H ? I strongly recommend the authors provide information on the T/S data set, a crucial aspect of the study.
The description of the steric estimate and the used T/S-data has been extended.

- Section 3.4, ocean mass component. This section is hard to follow, with several unclear sentences, e.g., lines 163-165.
Section has been rewritten is hopefully now more coherent to follow.

- Results section. I strongly recommend to put into perspective the results of this study with the published literature (e.g., the works by Henry et al, Armitage et al., Carret et al., Raj et al). What is novel ? Wht is different ? etc.
Results from other studies have been added where relevant throughout the results-section.

**Response to Interactive comment on* "Assessment of 21 years of Arctic Ocean Absolute Sea Level Trends (1995–2015)" *by* Carsten Ankjær Ludwigsen et al.**

**Anonymous Referee #2**
**Author response in red**

The points raised by the reviewer has been considered in the updated manuscript and the reviewer will hopefully find the manuscript improved.

When I reviewed the first submission of this manuscript, I recommended rejection. I cannot even remember if I have ever before made this recommendation and I have reviewed a great many manuscripts over the years. My assessment then was this: "this work reads like a technical report that has not properly matured into a scientific study, and I do not think that it merits publication". I now repeat this recommendation.

I started reading this second, revised, version, and I observe that the Introduction is very short (LL 16-43) and makes some (referenced) assertions about "uncertainties" (L 24) and "discrepancies" (LL 34-5), mentions that Arctic sea level reconstruction is "difficult" (L 31) and says that "satellite observations and in-situ observations are less consistent than in low and mid-latitudes"; but none of this provides a clear and explicit rationale and context for the work that is described in the rest of the manuscript.

The requirement on the author of any manuscript is to state in the Introduction (or Background, or Rationale) the following: (1) what is the "big issue", (2) what do we know, (3) what do we not know, (4) the reason for the importance of our ignorance, and (5) what we propose to do about it. If I remember right, the best-known (most accurately measured) quantity in physics is the electron magnetic moment, relative standard accuracy of 3x10^-10: everything is uncertain, and quantities in the environmental sciences may be very uncertain; but does that uncertainty matter? Is it important? Why? There is enough published in the Arctic context on altimetry, gravimetry, tide gauges, land movement (and so on) for the authors to fill in thoroughly on what text is needed to embrace all of these five points, and to make the case for why their work was worth doing. I strongly suspect that it was worth doing: but the authors still need to state their reasoning clearly, explicitly, and with evidence. A similar criticism applies to the Conclusions: what new have we learned, above and beyond what we already knew from previous studies? At present, the Conclusions are a summary but do not state what is new. This is why I stated in my first review that the manuscript resembled a technical report, and that assessment has not changed.

This manuscript is still not a mature document, and I consider it wrong first to send a document in such a condition to a journal, and then to expect the reviewers and the editor to repair its deficiencies. It is the authors' responsibility to think for themselves, to seek advice from colleagues, and to consult online resources, if that is what's required, in order to present a mature document for consideration for publication. Then the reviewers will engage willingly, so that it can be checked in the usual terms, and the process not used as a substitute for manuscript development.

---

## Author Response (AR3)

**Response to reviewer #1:**

In general my comments from the previous round of reviews have been addressed. There are still some minor grammar errors but, as stated last time, these should be captured at the proofing stage.

I do have two final queries on the manuscript:

Line 215: again, I would ask the authors to check that they do mean to refer to Armitage and Davidson (2014) and not Armitage et al (2016) here

Figure 7: I found it difficult to see the two different shades of green contours. While in general the lighter contours, for 95% confidence, fall within a large contour of 68% confidence, it can be hard in some places to tell if this is the case or if it is indicating a region that falls below 68% confidence. I would suggest making the light contour even lighter (or the 68% confidence a darker green) to help with this, or using an alternative colour such as magenta for the 95% confidence contour.

The two comments of reviewer #1 has been adressed and corrected.

**Response to reviewer #2:**

**General comment:**

This paper is not necessarily about GRACE. It is just that GRACE has limitations and that all other studies use GRACE for closing the sea level budget, while this paper choose a different approach that allows comparisons into the pre-GRACE era. The main concern about GRACE is that there is not one GRACE-solution, but GRACE-based mass estimates are constrained with a-priori knowledge which is interpreted differently across GRACE solutions. It is beyond the scope of this study to make an inter-GRACE comparison. Nevertheless, we have added two GRACE-products (JPL and GSFC RL06 mascons) to parts of the analysis.

BPR data are not useful for investigating secular mass-changes due to drift and changed geographic location (Proshutinsky et al, 2018 (JGR)). Trend estimates are very sensitive to small variations, so even though you could fit the monthly variation from GRACE to the BPR-record (like the 2010-paper of P-F), this would not guarantee that the BPR-trend is correct. Our results shows that GRACE doesn't add much information to understanding long-term sea level change in the Arctic and a theoretical approach (sea level fingerprints + IB) might as well be used (for long-term estimates). We have moderated the language accordingly at some places and don't claim that GRACE is worse than the approach used here. We just provide an approach to close the Arctic Sea Level Budget prior to 2003.

In my second review, I stated my dissatisfaction with the presentation of an immature manuscript and recommended rejection. I now review this third version, and I still do not find it to be publishable. Nevertheless, it is now at least in a reviewable form, so I will recommend "major revision" and state my reasons below.

The Abstract remains a list of things the authors have done; there is no statement of aim(s). Extension of record length is useful but not very interesting by itself. The authors have a lot

to say about the presumed deficiencies of GRACE products (fair enough) - so what, then, is their GRACE-related aim?

They generate new, separate records for changes in SSH due to steric changes and to mass distribution changes. The mass distribution changes rely fundamentally on tide gauge records. Tide gauge records are (by their nature) coastal. A good fit to coastal sea level is not evidence of good fit over the whole Arctic domain, however; at best it might be considered relevant to the shallow shelf seas. The reason for interest in GRACE in the Arctic is that it provides data on the deep, central part of the ocean. The authors cannot (as they do at the end of the manuscript) claim that their fields can validate GRACE when their fields are not validated away from the coast. There does exist a third resource apart from GRACE and their fields (independent, therefore) in the form of long-term bottom-pressure recorders in the Beaufort Gyre. They should use these (there may be more, maybe one in the North Pole observatory, I'm not sure about that) to check on their product in full ocean depth away from the coast.

There are two other good papers by Peralta-Ferriz (GRL 2010 and 2011) that have a lot of insightful material in this regard. Imagine a mode of SL variability that is like a drum resonance - pinned around the edge and moving up and down in the middle. Tide gauges do not see it. P-F and GRACE do. This is relevant to the Ekman spin-up / spin-down mechanism (Proshutinsky Phil Trans 2015, originally Proshutinsky & Johnson JGR 1997).

So what they need to do to make this manuscript acceptable is follow up on their introductory assertions about problems with GRACE near land and actually quantify the improvements they claim for their method by first comparing their fields with BPRs as an independent check, and then comparing the GRACE mass fields with theirs. GRACE products are readily available, so show the differences between their fields and GRACE. Is it true that they can demonstrate improvement near the coast? By how much? Is it significant? What about over the deep ocean where they tide gauge validation is not relevant? If their product is relatively unconstrained over the deep ocean, is GRACE actually better there?

Answer these questions satisfactorily and they may have an interesting, useful and properly validated study.

One detail: "However, the significant change in the Beaufort Sea coincides with the transition from Envisat to CryoSat-2 and a inter-satellite bias in DTU/TUM Altimetry can not be excluded". Armitage (JGR 2016) treats Envisat / Cryosat crossover in the supplementary material, so their statement is false.

Yes – but this claim is about the DTU/TUM sea level product and not Armitage (2016). Also, even though the mean fields (not directly crossovers) are subtracted for the overlap period, it doesn't necessarily mean that it removes the satellite bias when the overlap-period is happening in a time of rapid change (as in the Beaufort Sea in 2010).

---

## Author Response (AR4)

**Reviewer comment:** The authors are steadily improving the manuscript but I still cannot recommend it for publication. They really must validate their method in the deep ocean - the Beaufort Gyre, specifically - using BPRs. If they do not, then their approach is only very weakly validated, with correlations (R) <0.5, therefore $R^2$, the fraction of variance captured, is <0.25, which is not convincing.

They correctly observe in their replies to reviews that BPRs are not useful for long-term trends as a consequence of the well-known problem of instrument drift. However they *are* useful when de-trended for assessing seasonal cycles, as demonstrated by Peralta-Ferriz & Morison (GRL 2010). Now the biggest seasonal cycles in SSH & mass (from spin-up / spin-down) are seen in the Beaufort Gyre, and if their method can capture this seasonal cycle, then I will be convinced that the method has skill and that the paper is worth publishing.

We thank the reviewer for another review.

The reconstructed mass estimate is only reconstructed annually, since it is (partly) based on annual mass balance data from ice sheets and glaciers. Therefore, it is not meaningful to validate the reconstruction against seasonal BPR.

Instead, we have subtracted the steric estimate from altimetry and compared that with the BPR-record (and GRACE) in the Beaufort Sea, even though it feels that it is going away from the objective of this study to investigate contributions to *long term* sea level change in the Arctic. As you will see, the correlation to the BPR-record is quite good and better than the seasonal mean from GRACE.